# Density-Matrix Mean-Field Theory

Junyi Zhang[1, 2][*] and Zhengqian Cheng[3]

**1** William H. Miller III Department of Physics and Astronomy, Johns Hopkins University, Baltimore, Maryland 21218, USA
**2** Institute for Quantum Matter, Johns Hopkins University, Baltimore, Maryland, 21218, USA
**3** Department of Applied Physics and Applied Mathematics, Columbia University, New York, New York 10027, USA

[*] jzhan312@jhu.edu

## Abstract

Mean-field theories have proven to be efficient tools for exploring diverse phases of matter, complementing alternative methods that are more precise but also more computationally demanding. Conventional mean-field theories often fall short in capturing quantum fluctuations, which restricts their applicability to systems with significant quantum effects. In this article, we propose an improved mean-field theory, density-matrix mean-field theory (DMMFT). DMMFT constructs effective Hamiltonians, incorporating quantum environments shaped by entanglements, quantified by the reduced density matrices. Therefore, it offers a systematic and unbiased approach to account for the effects of fluctuations and entanglements in quantum ordered phases. As demonstrative examples, we show that DMMFT can not only quantitatively evaluate the renormalization of order parameters induced by quantum fluctuations, but can also detect the topological quantum phases. Additionally, we discuss the extensions of DMMFT for systems at finite temperatures and those with disorders. Our work provides an efficient approach to explore phases exhibiting unconventional quantum orders, which can be particularly beneficial for investigating frustrated spin systems in high spatial dimensions.

# 1   Introduction

Frustrated Hubbard and Heisenberg models [1–3] have continued to capture research attention over the last half-century due to their potential to host various intriguing quantum phases [4–8], as well as their relevance to high-$T_c$ superconductors [9] and their applications in quantum computations [10]. Determining the ground states of these models is often a challenging task.

Exact approaches frequently encounter limitations posed by the exponential wall. In exact diagonalization (ED), the dimension of the Hilbert space increases exponentially with the system size. In density-matrix renormalization group (DMRG) [11], while the area law provides relief from the exponential wall issue for gapped systems in one spatial dimension ($d = 1$), exponential scaling challenges remain for gapless systems or in higher dimensions ($d \geq 2$) [12–14]. On the other hand, quantum Monte Carlo (QMC) methods are less constrained by system size. Nevertheless, the notorious sign problem often plagues fermionic systems and frustrated magnetic systems [15, 16].

Approximation methods, serving as complements to exact approaches, prove to be useful and efficient tools for exploring various phases of a system. Conventional mean-field theories (MFTs) already provide insights into non-trivial effects arising from the interactions, such as the formation of local moments in metals [17], and the BCS theory for the superconductivity [18]. Conventional MFTs achieve simplification by neglecting the fluctuations; consequently, they tend to exhibit a bias towards ordered states and overlook nuanced effects stemming from the fluctuations.

Beyond conventional MFTs, various approximations have been proposed for fermionic systems [19–25]. In dynamical mean-field theory (DMFT), a lattice model is mapped to a local impurity model, and the effective action is constructed using Green's functions as dynamical mean fields [19]. This allows DMFT to capture the quantum features of the metal-insulator transitions [20]. While DMFT precisely describes systems in infinite dimensions, the computational demands of solving the local impurity problems with continuous baths necessitate ongoing efforts to simplify DMFT further. Recently, a quantum embedding method called density matrix embedding theory (DMET) has been introduced to enhance the efficiency of DMFT by taking the advantage of the frequency-independent local density matrix [21]. More recently, another simplification of DMFT known as variational discrete action theory (VDAT) ha been proposed. VDAT utilizes sequential product density matrices to variationally determine ground states [22–25].

Despite of the successes of DMFT and its simplifications for fermionic systems, a gap remains in methods beyond semiclassical mean-field approximations for spin systems. Although employing the Holstein-Primakoff transformation [26] allows spins to be mapped to bosons,

to which the DMFT may be adapted in principle, the transformation is nonlinear, and the semi-classical large-$S$ expansion becomes less controllable for $S = 1/2$ in the quantum limit, posing challenges for semiclassical methods applied to quantum spins. Similar challenges may be encountered with alternative methods. For instance, in the Schwinger boson representation [27], one needs to set $\mathcal{N} = 2$ for the quantum limit after a saddle-point mean-field approximation with large $\mathcal{N}$.

In this article, we propose a generalized mean-field method beyond conventional MFTs, which we call density-matrix mean-field theory (DMMFT). DMMFT constructs effective Hamiltonians, incorporating quantum environments shaped by entanglements quantified by reduced density matrices without presuming semiclassical orders. Therefore, it offers an unbiased approach to account for the effects of fluctuations and entanglements in quantum ordered phases. In contrast to QMC and DMFT, DMMFT is generically applicable to systems of fermions, bosons, as well as spins, regardless of frustrations. More importantly, by gauging the quantum fluctuations with the reduced DM, DMMFT can detect not only symmetry-breaking phases in Landau's paradigm but also topological phases with the help of entanglement spectra. Our work provides an efficient approach to explore phases that exhibit unconventional quantum orders. Particularly, it fills the gap left by the MFTs in studying quantum ordered phases in frustrated spin systems, where semiclassical methods become less controllable in the quantum limit, and QMC methods fail due to the sign problem.

Regarding DMMFT as a generalized cluster MFT, it is noteworthy that there are alternative cluster based methods used for studying quantum spin systems, such as the cluster variation method (CVM) and the linked-cluster expansion (LCE). Following Kikuchi's early study of the Ising model [28], CVM has also been extended for studying the quantum Heisenberg models [29–31]. CVM decomposes the entropic contribution to the free energy in terms of the cluster entropies that depends only on the reduced DMs over the clusters. It becomes precise when considering clusters up to the size of the entire system [32]. CVM provides a systematic framework for analyzing various cluster mean-field approximations, as demonstrated in Ref. 29. Examined from the perspective of CVM, DMMFT includes the high-order cluster corrections containing essential information of quantum fluctuations beyond conventional MFTs. Similarly, LCE also decomposes the corrections to physical observables into a series of terms depending on the clusters [33–38]. Early developments of the LCE treated the expansion of the free energy perturbatively in temperature, limiting its effectiveness to high temperatures [33–35]. Improvements have been made by extending LCE to calculate cluster physical observables non-perturbatively in temperature, also known as numerical LCE [36–38], which has been applied to some interesting frustrated quantum magnets [39,40]. Complementing LCE below the critical temperature, where long-range order develops, DMMFT does not suffer from convergence problems and can be more computationally efficient, as it does not require summation over a large number of clusters.

The rest of the article is organized as follows. In Sec. 2, we present the generic formulation of DMMFT. The mean-field equations of DMMFT are derived in parallel to those of conventional MFTs. Moreover, we demonstrate that DMMFT becomes equivalent to conventional MFTs when the hyperparameter that gauges quantum fluctuations is minimized. In Sec. 3, we apply the DMMFT to two demonstrative examples, 1) the Affleck-Khomoto-Lieb-Tasaki (AKLT) model [41–43], and 2) the antiferromagnetic Heisenberg model on triangular lattices (AFHTL) [44–58]. (Cf. Ref. 50 and references therein for a more comprehensive introduction to the frustrated Heisenberg models.) In the AKLT model, DMMFT is able to identify the topological ground states through their entanglement spectra. In the AFHTL, DMMFT reveals that quantum fluctuations not only renormalize the order parameters but also shift the phase boundaries. In Sec. 4, we compare DMMFT with DMRG and DMFT, as well as cluster-based methods CVM and LCE. Additionally, we discuss the extensions of DMMFT for systems at finite

temperatures and those with disorders. In Sec. 5, we draw conclusions and discuss potential avenues for future research. To aid readers in understanding and applying DMMFT, appendices formulating the iterative algorithm for solving DMMFT equations and providing pseudocodes implementing DMMFT for the AFHTL are presented in Append. A and B, respectively.

## 2 Density-Matrix Mean-Field Theory

In this section, we formulate DMMFT for a generic Hamiltonian. Let $\mathcal{I} = \{i\}$ denote the set of all sites. On each site, there is a collection of local operators $\mathcal{O}_i = \{O_i^\alpha\}$. A generic Hamiltonian composed of local operators can be organized as follows

$$H[\mathcal{O}_\mathcal{I}] = \sum_{c \in \mathcal{C}} H_c[\mathcal{O}_c] + \sum_{(c,c')} H_{c,c'}[\mathcal{O}_c, \mathcal{O}_{c'}] + \sum_{(c,c',c'')} H_{c,c',c''}[\mathcal{O}_c, \mathcal{O}_{c'}, \mathcal{O}_{c''}] + \ldots, \tag{1}$$

where $\mathcal{C} = \{c\}$ is a partition of $\mathcal{I}$, and $\mathcal{O}_\mathcal{S} = \cup_{i \in \mathcal{S}} \mathcal{O}_i$ is the collection of local operators over the sites within a set $\mathcal{S}$. In Eq.(1), $H_c$ depends only on the local operators within cluster $c$, while $H_{c,c'}, H_{c,c',c''}, \ldots$ describe the inter-cluster couplings. For systems with finite-range interactions, there exist proper partitions such that interactions involve only a finite number of clusters.

Conventional MFTs isolate a local cluster from the system and couple it to an effective environment, where the environment is assumed to be classical, and the correlated fluctuations between the cluster and the environment are neglected. DMMFT improves the conventional mean-field approximation by including the essential quantum fluctuations in the environment, where the reduced DM is used to gauge the quantum fluctuations and select Hilbert subspaces approximating the effective environment. In this section, we shall formulate DMMFT for the Hamiltonian in Eq.(1) without making additional assumptions about the specific local operators or the microscopic details of coupling terms. Therefore, DMMFT is a method generically applicable to fermions, bosons, as well as spins, irrespective of the presence of frustrations.

In the following subsections, we begin by reviewing the conventional mean-field approximation in Sec. 2.1. Then, we develop the mean-field equations and self-consistency conditions of DMMFT in parallel to those of conventional MFTs in Sec. 2.2. Following this, in Sec. 2.3, we conduct a comprehensive comparison between the DMMFT and the conventional MFTs. Within this comparison, we identify a hyperparameter, $n_c$, that interpolates the DMMFT and the conventional MFTs. Particularly, when $n_c = 1$, attaining its minimal value, DMMFT becomes equivalent to the conventional MFTs.

### 2.1 Conventional Mean-Field Approximation

We first review the approximations employed in conventional MFTs before delving into the development of DMMFT. In conventional MFTs, the mean-field decoupling localizes the operator products to individual Hilbert subspaces by neglecting the correlated fluctuations. More precisely, for an operator product $O_i^\alpha O_j^\beta$ acting on the Hilbert space $\mathcal{H}_i \otimes \mathcal{H}_j$, the conventional mean-field approximation decouples it as follwos

$$O_i^\alpha O_j^\beta \approx O_i^\alpha \langle O_j^\beta \rangle + \langle O_i^\alpha \rangle O_j^\beta - \langle O_i^\alpha \rangle \langle O_j^\beta \rangle, \tag{2}$$

where the terms on the right-hand side act on the subspace $\mathcal{H}_i$ or $\mathcal{H}_j$, or trivially as an additive $c$-number. Consequently, the correlated fluctuations $\langle \delta O_i^\alpha \delta O_j^\beta \rangle = \langle O_i^\alpha O_j^\beta \rangle - \langle O_i^\alpha \rangle \langle O_j^\beta \rangle$ vanish in this approximation. Alternatively, from the perspective of the quantum states, since the operator product factorizes in product states, $\langle \phi_i | \otimes \langle \phi_j | O_i^\alpha O_j^\beta | \phi_i \rangle \otimes | \phi_j \rangle = \langle \phi_i | O_i^\alpha | \phi_i \rangle \langle \phi_j | O_j^\beta | \phi_j \rangle$,

conventional MFTs implicitly assume the product structure of the states and neglect quantum entanglements.

Keeping this consideration in mind, we formulate the conventional mean-field approximation for the generic Hamiltonian in Eq.(1) as follows. Given a cluster $c$, let $\mathcal{C}'_c = \cup c'$ be the collection of clusters connected to $c$, referred to as the environment surrounding $c$. The associated Hilbert spaces are $\mathcal{H}_c$ and $\mathcal{H}_{\mathcal{C}'_c}$ for the cluster and the environment, respectively, where $\mathcal{H}_{\mathcal{S}} = \otimes_{i \in \mathcal{S}} \mathcal{H}_i$ for a set of sites $\mathcal{S}$. Retaining the terms within the extended cluster $\bar{c} = c \cup \mathcal{C}'_c$, the local Hamiltonian is

$$
\begin{aligned}
H[\mathcal{O}_{\bar{c}}] =& H_c[\mathcal{O}_c] + H_{c,\mathcal{C}'_c}[\mathcal{O}_c, \mathcal{O}_{\mathcal{C}'_c}] + H_{\mathcal{C}'_c}[\mathcal{O}_{\mathcal{C}'_c}], \\
H_{c,\mathcal{C}'_c}[\mathcal{O}_c, \mathcal{O}_{\mathcal{C}'_c}] =& \sum_{(c,c'):c'\in\mathcal{C}'_c} H_{c,c'}[\mathcal{O}_c, \mathcal{O}_{c'}] + \sum_{(c,c',c''):c',c''\in\mathcal{C}'_c} H_{c,c',c''}[\mathcal{O}_c, \mathcal{O}_{c'}, \mathcal{O}_{c''}] + \ldots, \\
H_{\mathcal{C}'_c}[\mathcal{O}_{\mathcal{C}'_c}] =& \sum_{c'\in\mathcal{C}'_c} H_{c'}[\mathcal{O}_{c'}] + \sum_{(c',c''):c',c''\in\mathcal{C}'_c} H_{c',c'}[\mathcal{O}_{c'}, \mathcal{O}_{c''}] + \ldots,
\end{aligned}
\tag{3}
$$

where $H_c$ is the Hamiltonian of the focused cluster $c$, $H_{\mathcal{C}'_c}$ is the Hamiltonian of the environment $\mathcal{C}'_c$, and $H_{c,\mathcal{C}'_c}$ represents the couplings between the cluster $c$ and its environment $\mathcal{C}'_c$. If the target state can be approximated by a product state locally, i.e.,

$$
|\phi_{\bar{c}}\rangle \approx |\phi_{\bar{c}}^{\mathrm{MF}}\rangle = |\phi_c\rangle \otimes |\phi_{\mathcal{C}'_c}\rangle,
\tag{4}
$$

a local effective Hamiltonian for the cluster $c$ can be obtained by substituting the operators over $\mathcal{H}_{\mathcal{C}'_c}$ with their expectation values in $|\phi_{\mathcal{C}'_c}\rangle$, i.e.,

$$
H_{\mathrm{MF}}^{(c)}[\mathcal{O}_c] = H_c[\mathcal{O}_c] + \langle\phi_{\mathcal{C}'_c}|H_{c,\mathcal{C}'_c}[\mathcal{O}_c, \mathcal{O}_{\mathcal{C}'_c}]|\phi_{\mathcal{C}'_c}\rangle + \langle\phi_{\mathcal{C}'_c}|H_{\mathcal{C}'_c}[\mathcal{O}_{\mathcal{C}'_c}]|\phi_{\mathcal{C}'_c}\rangle,
\tag{5}
$$

which acts only on the Hilbert subspace $\mathcal{H}_c$. One solves $|\phi_c\rangle$ as an eigenstate with respect to the effective Hamiltonian $H_{\mathrm{MF}}^{(c)}$ for each cluster $c$ locally. If one further assumes that the target state of the entire system can also be approximated by a product state, i.e.,

$$
|\Phi_{\mathrm{MF}}\rangle = \otimes_c |\phi_c\rangle,
\tag{6}
$$

then, for each extended cluster, $|\phi_{\mathcal{C}'_c}\rangle = \otimes_{c'\in\mathcal{C}'_c}|\phi_{c'}\rangle$. Therefore, Eq.(4), Eq.(5), and Eq.(6) form a closed set of coupled mean-field equations for conventional MFTs.

The mean-field equations can often be further simplified when systems have additional symmetries. If the partition $\mathcal{C} = \{c\}$ respects certain symmetries of the system, there exist symmetry transformations relating the clusters $T_{c',c} : c \mapsto c'$. A symmetric mean-field Ansatz state can be constructed simply as $|\Phi_{\mathrm{MF}}\rangle = \otimes_c T_{c,c_0}|\phi_{c_0}\rangle$. Particularly, for a translationally invariant state, all $|\phi_c\rangle$ can be chosen to be the same state $|\phi_{c_0}\rangle$ in the local Hilbert space $\mathcal{H}_{c_0}$. Thus, the coupled mean-field equations for the clusters reduce to a single set of self-consistent mean-field equations for $|\phi_{c_0}\rangle$.

The major assumption in the conventional MFTs, as described above, is that the target states can be approximated by product states [Eqs.(4) and (6)], which, nevertheless, is not justified *a priori*. For many ordered states, the correlated fluctuations arising from the quantum entanglements are not negligible, particularly at short range. Prototypically, in frustrated magnetic systems and symmetry-protected topologically ordered systems, such quantum fluctuations and quantum entanglements play essential roles. Therefore, improved treatments of quantum fluctuations and entanglements are necessary for a better understanding of emerging quantum ordered phases.

## 2.2 Density-Matrix Mean-Field Approximation

The simplifications achieved in conventional MFTs stem from the *separability*, which reduces the challenging task of studying the total Hamiltonian $H$ over an exponentially large Hilbert space $\mathcal{H}_{\mathcal{C}}$ to the more manageable task of studying the local Hamiltonians $H_{\mathrm{MF}}^{(c)}$ over local $\mathcal{H}_c$. However, it is not *necessary* for a system to be in a product state for two local subsystems to be separable. The assumptions of the product states [Eqs.(4) and (6)] can thus be relaxed. More precisely, for gapped systems, the correlated fluctuations $\langle \delta O_i^\alpha \delta O_j^\beta \rangle \to 0$ as $|i-j| \to \infty$. The absence of long-range entanglements ensures separability, allowing the local physics to be approximated with effective local systems. However, the short-range entanglements encode the quantum fluctuations, demanding a more faithful treatment.

Instead of using Eq.(2) for mean-field decoupling, it is instructive to recognize that the entanglement of a local cluster with an (infinite or finitely large) environment can always be *faithfully* reproduced within a finite extension of the cluster. More precisely, consider a generic state $|\Psi\rangle \in \mathcal{H}_{\mathcal{C}}$. The entanglement of the state $|\Psi\rangle$ over the cluster $c$ and the rest of the system $\mathcal{C}\backslash c$ can be characterized by the reduced DM

$$\rho_c = \mathrm{Tr}_{\mathcal{C}\backslash c} |\Psi\rangle\langle\Psi|. \tag{7}$$

The entanglement is controlled by $\mathcal{H}_c$ than $\mathcal{H}_{\mathcal{C}\backslash c}$ provided $D_{\mathcal{H}_c} < D_{\mathcal{H}_{\mathcal{C}\backslash c}}$, where $D_{\mathcal{H}}$ denotes the dimension of the Hilbert space $\mathcal{H}$. According to the purification theorem [59], there exists a state $|\tilde{\Psi}\rangle$ in $\tilde{\mathcal{H}} = \mathcal{H}_c \otimes \tilde{\mathcal{H}}_{\tilde{c}}$ such that $\mathrm{Tr}_{\tilde{c}} |\tilde{\Psi}\rangle\langle\tilde{\Psi}| = \tilde{\rho}_c$ is equivalent to $\rho_c$, and the dimension of the Hilbert subspace $\tilde{\mathcal{H}}_{\tilde{c}}$ is bounded by $\tilde{D}_c^E = \lceil \exp(SE_c) \rceil$, where

$$SE_c = -\mathrm{Tr}_c [\rho_c \ln(\rho_c)], \tag{8}$$

is the entanglement entropy and $\lceil q \rceil$ represents the smallest integer larger than or equal to $q$. In this context, DMMFT seeks for an effective Hamiltonian $\tilde{H}_{\mathrm{MF}}^{(c)}$ over some $\tilde{\mathcal{H}}$ (to be specified below), such that the reduced DM of the state obtained from the the Hamiltonian,

$$\rho_c^{\mathrm{MF}} = \mathrm{Tr}_{\tilde{c}} |\tilde{\phi}\rangle\langle\tilde{\phi}|, \tag{9}$$

well approximates the reduced DM of the target state $\rho_c = \mathrm{Tr}_{\mathcal{C}\backslash c} |\Phi\rangle\langle\Phi|$.

Instead of using Eq.(4), DMMFT assumes that *for systems without long-range entanglements, $\tilde{\mathcal{H}}$ can be found as a Hilbert subspace of $\mathcal{H}_{\tilde{c}}$ over the extended cluster $\tilde{c}$*. Mathematically, there exists a homomorphism

$$\Pi = \mathrm{Id}_c \otimes \Pi_{\mathcal{C}_c'} : \tilde{\mathcal{H}} \to \mathcal{H}_{\tilde{c}}, \tag{10}$$

and its pseudo-inverse $\Pi^\dagger : \mathcal{H}_{\tilde{c}} \to \tilde{\mathcal{H}}$ which acts as a projector.

To avoid confusion, it is essential to emphasize that the $\tilde{\mathcal{H}}$ constructed in DMMFT differs from that in DMET. The purification theorem only ensures the existence of $\tilde{\mathcal{H}}$ with its dimension bounded by $\tilde{D}_c^E$ from below. In DMET, an optimal $\tilde{\mathcal{H}}$ with the lowest possible dimension is employed, but this comes at the cost of a less straightforward construction of the embedding Hamiltonian. In contrast, DMMFT can intuitively construct an effective Hamiltonian by restricting the Hamiltonian for the extended cluster to $\tilde{\mathcal{H}}$, i.e.,

$$\tilde{H}_{\mathrm{MF}}^{(c)}[\mathcal{O}_c, \tilde{\mathcal{O}}_{\mathcal{C}_c'}] = \Pi^\dagger H[\mathcal{O}_{\tilde{c}}] \Pi, \tag{11}$$

where the local operators in $\tilde{\mathcal{O}}_{\mathcal{C}_c'}$ are given by

$$\tilde{O}_j^\beta = \Pi_{\mathcal{C}_c'}^\dagger O_j^\beta \Pi_{\mathcal{C}_c'}, \forall j \in \mathcal{C}_c'. \tag{12}$$

The reduced DM of the target state is approximated by $\rho_c^{\mathrm{MF}}$ [Eq.(9)] for an eigenstate $|\tilde{\phi}\rangle$ of $\tilde{H}_{\mathrm{MF}}^{(c)}$.

Reciprocally, for each $\rho_c$, we can construct the local projectors $\Pi_c$ as follows. Let $\left\{\lambda_i^{(c)}, |\lambda_i^{(c)}\rangle\right\}$ be the spectral decomposition of $\rho_c$, with the state vectors arranged in decreasing order according to their eigenvalues, i.e., $\lambda_i^{(c)} \geq \lambda_j^{(c)}, \forall i < j$. Define

$$\Pi_c = \sum_{i=1}^{n_c} |\lambda_i^{(c)}\rangle\langle\lambda_i^{(c)}|, \tag{13}$$

where $n_c \leq D_{\mathcal{H}_c}$ is a cut-off parameter. For the extended cluster $\bar{c} = c \cup \mathcal{C}_c'$, we choose

$$\Pi_{\mathcal{C}_c'} = \otimes_{c' \in \mathcal{C}_c'} \Pi_{c'}. \tag{14}$$

Parallel to the mean-field equations for the conventional MFTs, Eqs.(9) – (14) constitute a closed set of coupled mean-field equations for the DMMFT. In the presence of symmetries, the mean-field equations can be further simplified. Specifically, for a translationally invariant target state $|\Phi\rangle$, the reduced DMs of the local clusters are all identical to the same reduced DM $\rho_{c_0}$, and so are the projectors. Consequently, the coupled mean-field equations for the clusters reduce to a single set of self-consistent mean-field equations for $\rho_{c_0}$. The procedures for implementing the DMMFT algorithm are outlined in the Appendices.

## 2.3 Comparison of the Mean-Field Approximations

Comparing conventional MFT with DMMFT, we find that the cut-off parameter $n_c$ in Eq.(13) can be regarded as a hyperparameter interpolating between DMMFT and conventional MFT. To demonstrate this point, we first observe that when $D_{\mathcal{H}_{\mathcal{C}_c'}} = 1$, DMMFT becomes equivalent to a conventional MFT. If the ground state is a product state [Eq.(6)], the local reduced DM of cluster $c$ is $\rho_c = |\phi_c\rangle\langle\phi_c|$. The expectation values of the local observables agree

$$\langle O_i^\alpha \rangle_c = \langle\phi_c|O_i^\alpha|\phi_c\rangle = \mathrm{Tr}_c(\rho_c O_i^\alpha). \tag{15}$$

Moreover, since $\Pi = \mathrm{Id}_c \otimes (\otimes_{c'}|\phi_c\rangle\langle\phi_c|)$, $\tilde{\mathcal{H}}$ is isomorphic to $\mathcal{H}_c$, and the effective Hamiltonians $\tilde{H}_{\mathrm{MF}}^{(c)}$ are identical to $H_{\mathrm{MF}}^{(c)}$ under this isomorphism.

Furthermore, from Eq.(14), we have $D_{\mathcal{H}_{\mathcal{C}_c'}} = \prod_{c' \in \mathcal{C}_c'} n_{c'}$. Therefore, we may take $n_c$ as a hyperparameter interpolating between conventional MFT and DMMFT. When $n_c = 1$ (set to its minimal value), DMMFT simply reduces to conventional MFT, which underestimates the quantum fluctuations. When $n_c = D_{\mathcal{H}_c}$ (set to its maximal value), DMMFT describes a collection of (overlapped) extended cluster $\bar{c}$, which overestimate the quantum fluctuations compared to the infinite system in the thermodynamic limit. Thus, as $n_c$ increases from 1 to $D_{\mathcal{H}_c}$, an increasing amount of quantum fluctuations is included.

Since $n_c$ gauges the amount of quantum fluctuations included in DMMFT, ranging from underestimation to overestimation, it seems reasonable to expect an optimal value for $n_c$. This leads to the following *conjecture*.

**Conjecture.** *The optimal choice of the hyperparameters is given by*

$$n_c^* = \tilde{D}_c^E = \lceil \exp(SE_c) \rceil. \tag{16}$$

Although we do not have a mathematical proof for this conjecture, it does not undermine the practicality of DMMFT Particularly, one may use $n_c = \tilde{D}_c^E$ as a rule-of-thumb and treat $n_c$ as a *variational* hyperparameter.

It is easy to observe that $\rho_c$ generically has non-vanishing entanglement entropy whenever $D_{\mathcal{H}_{C'_c}} > 1$. This short-range entanglement is precisely what DMMFT captures beyond conventional MFTs. Consequently, DMMFT is expected to detect short-range entangled topological phases with a properly chosen $n_c$. In Sec. 3.1, we demonstrate with the AKLT model that DMMFT can indeed detect the topological ground states through the entanglement spectra without being biased to the symmetry-breaking states.

Lastly, we comment on the separability in DMMFT. In contrast to Eq.(6) used in conventional MFTs, DMMFT does not assume the product structure of the state. Instead, DMMFT adopts a weaker form of separability characterized by

$$\rho^{(2)}_{c_1,c_2} = \text{Tr}_{C \backslash (c_1 \cup c_2)} |\Phi\rangle\langle\Phi| \approx \rho_{c_1} \otimes \rho_{c_2}, \tag{17}$$

for any two clusters $c_1$ and $c_2$ that are not connected.

## 3 Applications

In this section, we apply the DMMFT to two demonstrative examples, the AKLT model and the AFHTL, that have been extensively studied in the literature. The AKLT model was historically proposed to confirm Haldane's conjecture regarding antiferromagnetic spin-1 chains [41–43]. One notable feature of the AKLT model is that its ground state has a closed analytic form. Nevertheless, we choose the AKLT model as our illustrative example not solely because of its known ground state wave function, but primarily due to its intriguing property that the expectations of local moments, and consequently the usual Néel order parameter (alternating magnetization), vanish in the ground state [42,43]. This property challenges the ability of conventional MFTs to detect the orders borne in the ground states of the AKLT model. Moreover, despite proposals suggesting that topological orders in one dimension can be characterized by nonlocal order parameters, such as string order [60,61], it is still desirable to identify local features that can distinguish topological ordered states from disordered or topologically trivial states, especially at the mean-field level. In Sec. 3.1, we demonstrate that DMMFT can correctly detect the topological ground states based on the reduced DM and the entanglement spectra. All local observables that can serve as order parameter can be straightforwardly calculated with respect to the reduced DM according to Eq.(15). Nevertheless, the entanglement spectra reveal additional information beyond conventional MFTs, reflecting the quantum nature of orders. It is important to emphasize that the order in the ground states of the AKLT model is a generic feature of this exotic topological phase, which is not restricted to the special point of the AKLT model, but persists upon perturbations that do not close the gap. The exact ground state wave function of the AKLT model is used here only as a convenient benchmark for quantitative comparisons.

The AFHTL serves as a prototypical model for studying frustrated magnetism. Inspired by Fazekas and Anderson's pioneering study [5], various intriguing phases have been proposed to potentially exist in the AFHTL due to its frustrations. [6,14,62–68]. In Sec. 3.2, we apply the DMMFT to the simplest case of the AFHTL, which includes only nearest-neighbor couplings and easy-axis anisotropy. Even in this simplest case, the ground state of the system exhibits a noncollinear three-sublattice order, reminiscent of the semiclassical Néel state. However, the order parameters are subjected to significant renormalization due to the cooperative interactions between quantum fluctuations and frustration-induced noncollinearity [44–46]. Additionally, we report on the magnetization of the AFHTL in a longitudinal magnetic field, which has been extensively studied in both classical and quantum contexts [47–57]. Notably, quantum fluctuations play a non-trivial role in stabilizing the ordered states in the AFHTL [56–58]. For quantitative comparison, we compute the magnetization curve using the state-of-the-art technique,

DMRG. Our DMMFT results show good agreement with DMRG. It is important to emphasize that the simple AFHTL, with only nearest-neighbor couplings and easy-axis anisotropy in a longitudinal external magnetic field, serves as a robust benchmark because it has been extensively studied using various methods and confirmed in the literature. Nevertheless, DMMFT is not limited to the simple AFHTL. It is capable of detecting stripe phases in the more complicated $J_1$-$J_2$ AFHTL [14], as well as resonating valence bond phases with dimerized spins [5]. We would highlight DMMFT as a generic and unbiased approach to characterizing diverse phases with both classical and quantum orders.

## 3.1 Affleck-Kennedy-Lieb-Tasaki Model

We apply the DMMFT to the AKLT model. The AKLT model is a one-dimensional spin-1 chain defined by the Hamiltonian

$$H_{\text{AKLT}} = \sum_i P_{i,i+1}^{(S=2)} = \sum_i \frac{1}{2}\big[\mathbf{S}_i \cdot \mathbf{S}_{i+1} + \beta(\mathbf{S}_i \cdot \mathbf{S}_{i+1})^2 + \gamma\big], \tag{18}$$

where $\beta$ describes the biquadratic coupling and $\gamma$ is an additive constant. The operators $\mathbf{S}_i$ are spin-1 operators with representations in the $S^z$-basis given by

$$S_i^x = \frac{1}{\sqrt{2}}\begin{pmatrix} 0 & 1 & 0 \\ 1 & 0 & 1 \\ 0 & 1 & 0 \end{pmatrix}, S_i^y = \frac{1}{\sqrt{2}}\begin{pmatrix} 0 & -i & 0 \\ i & 0 & -i \\ 0 & i & 0 \end{pmatrix}, S_i^z = \begin{pmatrix} 1 & 0 & 0 \\ 0 & 0 & 0 \\ 0 & 0 & -1 \end{pmatrix}. \tag{19}$$

The coefficients $\beta$ and $\gamma$ are determined by the condition that each term in the Hamiltonian is a projector to the Hilbert subspace of total spin $S = 2$. More precisely, when $\beta = 1/3$, and $\gamma = 2/3$,

$$P_{i,i+1}^{(S=2)}(x) = \frac{1}{2}(x + \beta x^2 + \gamma) = \begin{cases} 1, & x = 1, \\ 0, & x = -1, -2. \end{cases} \tag{20}$$

where $x = \mathbf{S}_i \cdot \mathbf{S}_{i+1} = 1, -1, -2$ in total spin sectors $\mathscr{H}_{\{i,i+1\}}^{(S=2,1,0)}$ respectively. By construction, $H_{\text{AKLT}}$ describes antiferromagnetic nearest neighbor couplings, penalizing neighboring spins only when they are in $S = 2$ states.

The ground state of the AKLT model is exactly known and showcases non-trivial topological order. Notably, it features spin-1/2 edge states (fractional to spin-1) and exhibits a four-fold degeneracy for an open chain [43]. This topological characteristic persists even for $\beta < \frac{1}{3}$ deviating from the special point of the AKLT model, provided the gap does not close [60, 61]. Moreover, the Néel order vanishes in the ground state, rendering conventional MFTs ineffective in distinguishing it from a trivial paramagnetic phase. Nevertheless, by utilizing reduced DMs and entanglement spectra, DMMFT effectively identifies the topological ground state.

Let us consider a two-site cluster $c$ located in the bulk of an infinite chain. By translational symmetry, any two-site cluster $c$ within the chain shares the same reduced DM $\rho_c^{(2)}$. The spectrum of this two-site reduced DM is exactly known and given by $\text{eig}\rho_c^{(2)} = \left\{\frac{1}{3}, \frac{2}{9}, \frac{2}{9}, \frac{2}{9}, 0, 0, 0, 0, 0\right\}$. Furthermore, upon tracing out one of the sites within the cluster, we obtain the single-site reduced DM whose spectrum is exactly known, $\text{eig}\rho_c^{(1)} = \left\{\frac{1}{3}, \frac{1}{3}, \frac{1}{3}\right\}$.

We apply the DMMFT to the AKLT model, focusing on the two-site cluster $c$ and solving the reduced DM $\rho_c^{(2)}$ self-consistently using the mean-field equations as described in Sec. 2.2. Specifically, the reduced DM is computed over an extended cluster that includes the left and right nearest-neighbor two-site clusters adjacent to $c$. The associated Hilbert space is denoted as $\tilde{\mathscr{H}}_{\tilde{c}} = \tilde{\mathscr{H}}_l \otimes \mathscr{H}_c \otimes \tilde{\mathscr{H}}_r$, where $\tilde{\mathscr{H}}_{l,r} \sim \Pi_c(\mathscr{H}_c)$ are the Hilbert subspaces selected by the reduced DM, as defined in Eq.(13). We set the cut-off parameter $n_c = 4$ for our calculations.

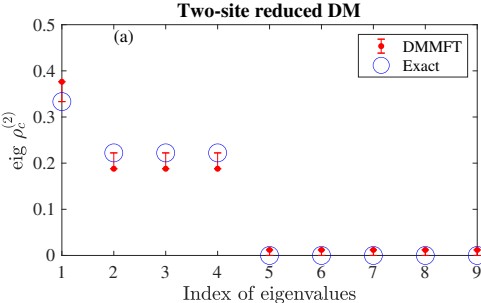
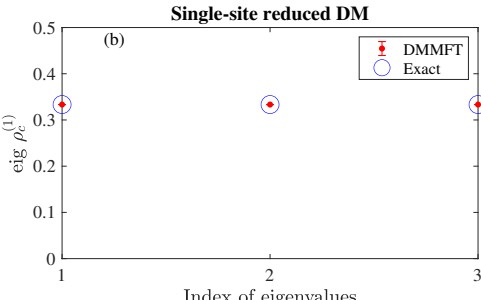

Figure 1: Spectra of the two-site and single-site reduced DMs for the AKLT model. The eigenvalues $\{\lambda_i^{(n)}\}$ of the reduced DM $\rho_c^{(n)}$, indexed by $i$ in decreasing order, are plotted. Blue circles represent the exact values, and red dots denote the DMMFT results. Error bars are used to indicate the deviations of the DMMFT results from the exact values.

In Fig. 1, the spectra of the two-site and single-site reduced DMs are presented. Blue circles represent the exact values, while red dots depict the results obtained using DMMFT. The spectrum of the two-site reduced DM obtained with the DMMFT reasonably aligns with the exact values [Fig. 1(a)], notably capturing the correct degeneracies. The entanglement entropy of the two-site cluster evaluated by DMMFT is $SE_{c,\mathrm{MF}}^{(2)} = 1.58$. Although it is slightly larger than the exact value $SE_{c,\mathrm{exact}}^{(2)} = 1.3689$, this discrepancy is expected due to the cut-off parameter $n_c$ chosen, which is marginally larger than $\exp(SE_{c,\mathrm{exact}}^{(2)}) = 3.9310$. Fig. 1(b) displays the spectrum of the single-site reduced DM, where the DMMFT results align excellently with the exact values. The single-site entanglement entropy $SE_c^{(1)} = 1.0986 = \ln(3)$ also matches. Furthermore, the expectation values of two spins within the cluster $\langle S_{1,2}^\alpha \rangle = \mathrm{Tr}_c(S_{1,2}^\alpha \rho_c)$ both vanish. This indicates that the ground states found by DMMFT are not biased towards the semiclassical Néel states, which is consistent with the exact results.

## 3.2 Antiferromagnetic Heisenberg Model on Triangular Lattices

We apply the DMMFT to another illustrative example, the simple AFHTL. The Hamiltonian for the AFHTL including nearest-neighbor anisotropic exchange interactions is given by

$$
\begin{aligned}
H_{AFH} &= \sum_{(i,j)} \Big[ J_{xy} \left( S_i^x S_j^x + S_i^y S_j^y \right) + J_z S_i^z S_j^z \Big] \\
&= \sum_{(i,j)} J \Big[ \mathcal{A} \left( S_i^x S_j^x + S_i^y S_j^y \right) + S_i^z S_j^z \Big],
\end{aligned}
\tag{21}
$$

where $J_z$ and $J_{xy}$ denote the strengths of longitudinal and transverse exchange interactions, respectively. For antiferromagnetic couplings, the model is more conveniently parameterized with $J = J_z > 0$ and $\mathcal{A} = J_{xy}/J_z$. The phases of the system depend solely on the dimensionless parameter $\mathcal{A}$ that specifies the exchange anisotropy. Our focus is on the easy-axis case, i.e., $0 \leq \mathcal{A} \leq 1$. The two extreme values of $\mathcal{A}$ are: 1) $\mathcal{A} = 0$, where the transverse exchange terms vanish (often referred to as the Ising limit), and 2) $\mathcal{A} = 1$, where the exchange interactions exhibit full rotational symmetry (often referred to as the Heisenberg limit). Both limits have been extensively investigated in the literature. Particularly, in the Ising limit, quantum fluctuations are additionally introduced by a transverse magnetic field, and the sign problem of the diagonal (longitudinal) frustrations can be mitigated in QMC simulations [58, 69]. On the

other hand, in the Heisenberg limit, the system orders in the so-called 120°-Néel state, which has been confirmed by various numerical methods [44–46].

While the phase diagram of the AFHTL in a longitudinal external magnetic field is generally bounded by the Ising and Heisenberg limits for a generic anisotropy parameter $0 < \mathcal{A} < 1$, detailed computations of its quantum phases remain challenging, particularly due to the QMC sign problem when $\mathcal{A} \neq 0$. Previous studies, such as those in Refs. 52, 53, have examined the magnetization using large-size cluster mean-field theory (belonging to the class of conventional MFTs) with a scaling scheme. Additionally, Ref. 47 and Ref. 49 have reported studies of the magnetization using ED, and Ref. 48 has presented a phase diagram obtained with DMRG. These previous investigations provide comprehensive insights into the behavior of the AFHTL in a magnetic field and serve as valuable benchmarks for understanding its quantum phase diagram.

Below, we present our investigation of the AFHTL using DMMFT. To benchmark our findings, we also performed a calculation using the state-of-the-art technique, DMRG. Our results obtained with DMMFT are compared both to DMRG calculations and to results reported in the literature. Remarkably, we observe quantitative agreement between the DMMFT results and those obtained from more sophisticated methods. Notably, this alignment is achieved using a small cluster of just three lattice sites in DMMFT, underscoring its efficiency and accuracy in characterizing the AFHTL.

The Hamiltonian describing the AFHTL in a longitudinal external field includes both exchange interactions and Zeeman couplings,

$$H_{\text{AFH,Z}} = H_{\text{AFH}} + H_Z, \tag{22}$$

where $H_Z = -\sum_i g\mu_B h_z S_i^z$. In our DMMFT calculations, we fix the gyromagnetic factor as $g\mu_B = 1$ and $h_z$ is measured in the corresponding natural units.

Four distinct phases emerge in the AFHTL with generic easy-axis anisotropy under a longitudinal external magnetic field. These phases are the coplanar "Y"-shaped phase, the collinear up-up-down (UUD) phase, the coplanar "V"-shaped phase, and the collinear polarized phase, which appear in increasing order of $h_z$. All these phases are compatible with the partitioning of the lattice into three-site clusters and with the translational symmetries inherent to these clusters. A simple order parameter distinguishing these phases is the magnetization along the easy axis, i.e.,

$$M_c^z = \sum_{i \in c} S_i^z = S_A^z + S_B^z + S_C^z, \tag{23}$$

where the subscript $i = A, B, C$ labels the three sublattice sites within the cluster $c$. The polarized and UUD phases are characterized by magnetization plateaux at $M_c^z = M_{c,(s)}^z$ and $M_c^z = M_{c,(s)}^z / 3$, respectively, where $M_{c,(s)}^z = \frac{3}{2}$ is the saturation magnetization of the cluster. The "V"-shaped phase lies between the UUD and polarized phases, while the "Y"-shaped phase occurs at low fields. Another distinguishing order parameter characterizing the non-collinearity of the "Y"-shaped phase is the vector chirality,

$$\boldsymbol{\kappa}_V = \frac{8}{3\sqrt{3}} \left( \mathbf{S}_A \times \mathbf{S}_B + \mathbf{S}_B \times \mathbf{S}_C + \mathbf{S}_C \times \mathbf{S}_A \right), \tag{24}$$

where the normalization factor is chosen such that $|\boldsymbol{\kappa}_V| = 1$ for the 120°-Néel state of $S = \frac{1}{2}$ spins without being renormalized by quantum fluctuations.

Despite that all four phases are homologous to their corresponding semiclassical Néel ordered states, quantum fluctuations play crucial roles in the "Y"-shaped and UUD phases. Notably, quantum fluctuations lift the accidental degeneracies within the semiclassical ground

state manifold, a phenomenon known as quantum order-by-disorder [56,57]. Moreover, these fluctuations significantly renormalize both the magnitudes of the ordered spins and the corresponding order parameters [44–46].

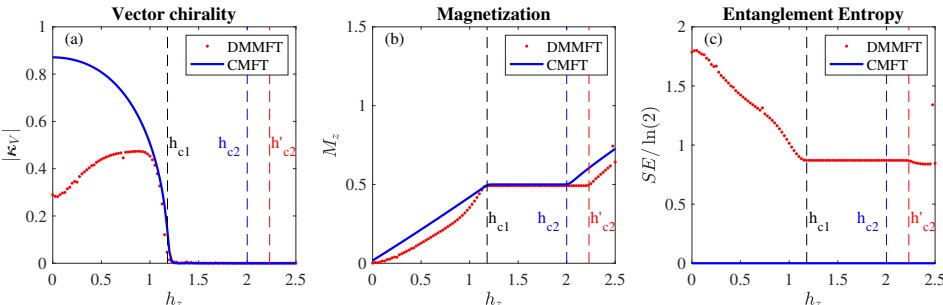

Figure 2: Phase diagram of AFHTL in a longitudinal magnetic field ($\mathcal{A} = 0.9$). (a) Vector chirality. (b) Magnetization. (c) Entanglement entropy. Red dots represent DMMFT results, while blue curves depict conventional mean-field theory (CMFT) results. Vertical dashed lines indicate phase boundaries. The critical field $h_{c1}$ distinguishes the "Y"-shaped phase from the UUD phase, where DMMFT and CMFT agree. The critical field $h_{c2}$ distinguishes the UUD phase from the "V"-shaped phase. Determined from the magnetization curves, the value of $h_{c2}$ (blue) from CMFT exhibits a discernible difference compared to $h'_{c2}$ (red) from DMMFT.

In our DMMFT calculations, we select an extended cluster $\bar{c}$ comprising of four three-lattice clusters arranged with periodic boundary conditions. The use of periodic boundary conditions helps minimize boundary effects, as the Hilbert subspaces selected by the reduced DM are sensitive to boundary conditions, as observed in DMRG studies [70]. The geometry and implementation details of DMMFT for the AFHTL are elaborated in Append. B, where pseudocode is also provided.

In Fig. 2, we present the dependence of vector chirality, magnetization, and entanglement entropy of the ground states of AFHTL (with anisotropy parameter $\mathcal{A} = 0.9$) on the longitudinal magnetic field. The results are calculated using DMMFT (red) and the conventional MFT (blue). The cut-off parameter $n_c$, which interpolates between conventional MFT and DMMFT as discussed in Sec. 2.3, is set to $n_c = 4$ for DMMFT and $n_c = 1$ for conventional MFT.

In Fig. 2(a), the vector chirality calculated using conventional MFT decreases monotonically as $h_z$ increases in the "Y"-shaped phase and vanishes in the UUD and "V"-shaped phases. In sharp contrast, the vector chirality calculated using DMMFT does not decrease monotonically in the "Y"-shaped phase. Notably, the vector chirality at $h_z = 0$ in DMMFT is only about 30% of that in conventional MFT. The discrepancy reflects the renormalization of the ordered spins due to the quantum fluctuations. Upon closer examination of the magnitude of the ordered spins, for a small anisotropy $\mathcal{A} = 0.9$, we find $|\langle \mathbf{S} \rangle| = 0.269 \pm 0.014$. This value, only about 50% of $S = \frac{1}{2}$, aligns with expectations based on previously reported values [44–46]. Given that $\boldsymbol{\kappa}_V$ scales as $S^2$, a reduction factor of approximately 25% is expected for the magnitude of the vector chirality at $h_z = 0$. The curves of DMMFT and conventional MFT converge as $h_z$ increases, both vanishing beyond $h_{c1}$ (the critical field separating the "Y"-shaped and UUD phases, determined from the magnetization curves). With stronger $h_z$, the spins align closer to the easy axis, and the effect of quantum fluctuations weakens, which is clearly evidenced by the monotonically decreasing entanglement entropy in the "Y"-shaped phase as shown in Fig. 2(c). Thus, the suppression of the quantum fluctuations and the alignment of the spins towards the collinear UUD configuration by the longitudinal magnetic field jointly lead to the non-monotonic dependence of the vector chirality $|\boldsymbol{\kappa}_V|$ on $h_z$.

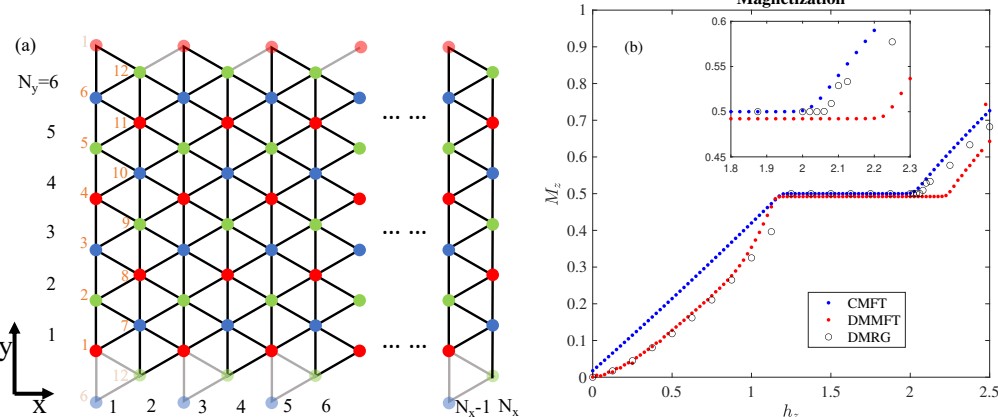

Figure 3: (a) Cylindrical geometry YC$N_y$-$N_x$ of a triangular lattice used for DMRG calculations. Translucent sites and bonds represent the extended triangular lattice under the periodic boundary condition. Sites in the tensor string are labeled by numbers in orange. (b) Magnetization curve of the AFHTL in longitudinal magnetic field ($\mathcal{A} = 0.9$). The results from CMFT (blue dots), DMMFT (red dots), and DMRG (black circles) are overlaid. The inset provides a zoomed-in view of the transition between the UUD phase and the "V"-shaped phase.

In Fig. 2 (b), notable differences are evident in the magnetization curves predicted by DMMFT and conventional MFT. Specifically, DMMFT shows a nonlinear magnetization curve at low fields in the "Y"-shaped phase and a wider plateau supporting the UUD phase. To benchmark our results, we also perform a calculation of the magnetization using the state-of-the-art technique DMRG [71]. We consider a triangular lattice with $N_x \times N_y$ sites, as shown in Fig. 3(a). We impose the periodic boundary condition along $y$-direction, effectively wrapping the triangular lattice into a cylinder with its axis parallel to the $x$-direction. (The translucent sites and bonds represent the extended triangular lattice under the periodic boundary condition to aid visualization.) Since one edge of the triangular lattice is parallel to the $y$-direction, we refer to this geometry as YC$N_y$-$N_x$, following the convention used in the literatures [62, 63, 66, 67], where $N_y$ is the circumference, and $N_x$ is the length of the cylinder. A tensor string winds through the system helically as $i_s = N_y(n_x - 1) + n_y$, where $i_s$ labels the site in the tensor string [shown as orange numbers in Fig. 3(a)], and $n_x$ and $n_y$ are integer coordinates of the sites in the $x$- and $y$-directions.

Fig. 3(b) shows the magnetization curve calculated with DMRG for YC6-30 and bond dimension of $D_b = 400$ (black circles), overlaid with the results of conventional MFT (blue dots) and DMMFT (red dots). The DMRG results are in excellent agreement with the DMMFT results, confirming the nonlinear magnetization at low field. Additionally, a self-consistency check for the choice of $n_c = 4$ in DMMFT can be confirmed from the entanglement entropy in the "Y"-shaped phase. At $h_z = 0$, we have $SE_c \approx 1.8\ln(2)$. Therefore, $\tilde{D}_c^E = \lceil \exp(SE_c) \rceil = 4$.

On the other hand, as $h_z$ surpasses $h_{c2}$, the system transitions towards the "V"-shaped phase, deviating from the $\frac{1}{3}$ magnetization plateau of the UUD phase. Fig. 2(b) shows a discernible difference in the critical field determined from the magnetization curves in DMMFT ($h'_{c2} \approx 2.2$) and in conventional MFT ($h_{c2} \approx 2.0$). Notably, the field range stabilizing the UUD phase is larger in DMMFT compared to conventional MFT. Although the UUD configuration is homologous to its semiclassical counterpart, quantum fluctuations contribute significantly to stabilize the UUD phase [52, 54]. Especially in the Heisenberg limit, the UUD phase is stable

only $h_z = h_{c2} = \frac{1}{3}h_z^{(s)} = 1.5$ (where $h_z^{(s)}$ is the saturation field) in the classical phase diagram. However, it extends to a finite field range due to quantum fluctuations in the quantum phase diagram. A similar effect of quantum fluctuations is also anticipated when $\mathcal{A} < 1$. Since $n_c$ used in DMMFT is larger than that in conventional MFT, incorporating more quantum fluctuations, a larger field range ($h'_{c2} > h_{c2}$) that stabilizes the UUD phase is expected.

To provide a quantitative comparison, we further compare the value of $h_{c2}$ obtained from DMMFT to those estimated from our DMRG results and those inferred from the literature. The inset of Fig. 3(b) provides a zoomed-in view of the transition between the UUD phase and the "V"-shaped phase. Our DMRG results indicate $h^*_{c2} \approx 2.06$, which falls between the $h_{c2}$ of conventional MFT and $h'_{c2}$ of DMMFT. Few precise values of $h_{c2}$ for $\mathcal{A} = 0.9$ have been reported in the literature [47–49, 52, 53]. Inferring from the reported values of $h_{c2}$ for the isotropic Heisenberg model, we conclude that the $h^*_{c2}$ estimated from our DMRG results may slightly underestimate the critical field, probably due to the finite-size effect of $N_y = 6$ along the transverse direction of the cylinder [compared to Fig. 1 in Ref. 48 and Fig. 7(a) in Ref. 53 for scaling to thermodynamic limit]. Conversely, $h'_{c2}$ estimated from our DMMFT results may slightly overestimate the critical field due to our choice of $n_c = 4$, which is larger than the speculated optimal $n^*_c = 2$ estimated from the entanglement entropy. This choice includes too many quantum fluctuations, favoring the UUD phase. Moreover, as $h_z$ drives the system deeper into the "V"-shaped phase, the magnetization curve obtained from our DMRG calculation gradually converges towards that of DMMFT.

In conclusion, our study of the AFHTL demonstrates the systematic improvements achieved by DMMFT over conventional MFT. Remarkably, even with a small cluster of just three lattice sites, DMMFT attains precision comparable to that of large system sizes calculated using DMRG or conventional MFT plus scaling. This underscores the efficiency and effectiveness of DMMFT in capturing the effects of quantum fluctuations.

# 4    Discussions

In Sec. 4.1, we conduct a comparative analysis of DMMFT with DMRG, DMFT, CVM, and LCE. Additionally, in Sec. 4.2, we extend the DMMFT for systems at finite temperatures, and in Sec. 4.3, we extend the DMMFT for systems with disorders.

## 4.1    Comparisons with Alternative Methods

### 4.1.1    Comparison with Density-Matrix Renormalization Group

We first compare DMMFT with DMRG. The construction of the effective Hamiltonian in both methods appears similar, as both use the reduced DM to select a significant Hilbert subspace with respect to the target state. This resemblance between the DMMFT and the infinite DMRG is most pronounced when the spatial dimension $d = 1$, as illustrated in the AKLT model in Sec. 3.1. The primary difference lies in where the Hilbert space $\tilde{\mathcal{H}} = \mathcal{H}_c \otimes \tilde{\mathcal{H}}_{\tilde{c}}$ is chosen. In DMMFT, the projector $\Pi_{\mathcal{C}'_c}$ is iteratively optimized solely over the Hilbert space $\mathcal{H}_{\mathcal{C}'_c}$ without changing the cluster size of the environment. Conversely, in infinite DMRG, the environment is constructed iteratively over the Hilbert space of the semi-infinite chains. When gauged by quantum entanglement, the dimension of the Hilbert subspace selected for constructing the effective Hamiltonian is bounded by $\tilde{D}^E_c$ [Eq.(16)] in both cases. Since $\tilde{D}^E_c$ is bounded by the size of the cluster $c$, DMRG can feasibly select a finite significant Hilbert subspace without being hindered by the infinite size of the environment asymptotically. In contrast, DMMFT assumes that the significant Hilbert subspace resides within $\mathcal{H}_{\mathcal{C}'_c}$, which is a reasonable approximation, especially for short-range entangled systems. Disregarding practical constraints and assuming

the chosen cluster size is larger than the typical entanglement range, DMMFT and DMRG are equivalent in $d = 1$.

The advantages of DMMFT become more significant in higher dimensions, $d \geq 2$. In DMRG, a $d$-dimensional system is compactified into a quasi-one-dimensional system, where an artificial one-dimensional string of clusters winds through the system, as illustrated in Fig. 3(a). Consequently, even for short-range entangled systems, artificial long-range entanglements emerge for sites that are physically close but distant along the string. The entanglement between physically neighboring clusters scales exponentially with $N_\perp^{d-1}$, where $N_\perp$ represents the linear scale of the system along the transverse directions. In contrast, DMMFT makes use of the separability [Eq.(17)] to study local problems over extended clusters without introducing artificial long-range entanglements arising from the transverse dimensions. From this perspective, DMMFT can be viewed as a mean-field approximation of DMRG and proves particularly advantageous for studying short-range entangled systems in higher dimensions.

### 4.1.2 Comparison with Dynamical Mean-Field Theory

We proceed to compare DMMFT with DMFT. In DMFT, designed for fermionic systems, the effective action is constructed over a local cluster, using the single-particle Green's function as a dynamical mean-field to capture the fluctuations [19,20]. This local problem, entailing a dynamical mean-field, is subsequently mapped to an effective impurity model and solved using ED or QMC kernels [19]. Unlike the Anderson impurity model tailored for fermions, devising an equivalent impurity model for spins is not straightforward. Furthermore, even if a spin impurity model consistent with the principles of DMFT could be formulated, efficiently solving it would remain challenging. Constructing a generic spin impurity (if not impossible) can introduce significant frustrations, which leads to the failure of the QMC kernel due to the sign problem. Additionally, achieving an accurate representation of the environment requires incorporating a large number of spins beyond those in the local cluster, which causes the ED kernel to rapidly encounter the exponential wall. A simplification of DMFT proposed in DMET for fermionic systems involves utilizing the local density matrix, a static observable, instead of the dynamical Green's function [21]. While a formal generalization of DMET to spin systems is feasible, the challenge lies in identifying a suitable reference wave-function that facilitates straightforward Schmidt decomposition and the construction of the embedding Hamiltonian. In contrast, in DMMFT, the reduced DM is naturally defined for many-body states, making DMMFT generally applicable to fermions, bosons, as well as spins.

An alternative approach to investigating the ground state of a quantum many-body system is to work directly with the wave function. One example is the Gutzwiller method, which uses Gutzwiller wave function as a variational Ansatz [72–75]. The Gutzwiller approximation can be viewed as a quantum embedding approach in a unified perspective with DMFT and DMET [76]. More recently, the VDAT has been proposed to address the shortcomings of the Gutzwiller approximation and simplify the computational complexity of DMFT. In VDAT, the variational Ansatz is the sequential product density matrix, evaluated via the discrete action theory [22,23]. The VDAT Ansatz is controlled by an integral parameter $\mathcal{N}$, and it is asymptotically exact as $\mathcal{N}$ tends to infinity, although it also becomes more computationally demanding. VDAT has been shown to yield accurate ground state wave functions for fermionic systems, even with $\mathcal{N}$ as small as 3 [24,25]. However, it faces challenges when applied to spin systems. While constructing the sequential product density matrix for the spin system is straightforward, the absence of Wick's theorem for spin systems presents a challenge in efficiently applying discrete action theory to evaluate the Ansatz.

Moreover, integrating wave function approaches with DMMFT raises the question of determining a global state from reduced density matrices of local clusters. This challenge is recognized as the quantum marginal problem [77]. Finding a general solution to this problem

is currently beyond the scope of this paper and remains an open and challenging research question [78].

It is particularly important to address the issue of lattice symmetries when comparing DMMFT with cluster extensions of DMFT. The problem of maintaining lattice symmetry is shared not only by DMFT but by all cluster approaches formulated in real space. In cluster extensions of DMFT, this challenge arises because the approximate self-energy might not preserve the underlying lattice symmetry, leading to inconsistencies if artificial periodization is not appropriately imposed [79]. In contrast, DMMFT works with local static Hamiltonians, which allows for a more straightforward imposition of symmetrization. Let $G = g$ denote a subgroup of symmetry transformations of the lattice that one would enforce, and denote the representations of $g$ over $\tilde{\mathcal{H}}$ as $\tilde{g}$. Then, the mean-field Hamiltonian in Eq.(11) can be symmetrized as

$$\tilde{H}_{\text{MF,Symm}}^{(c)} = \frac{1}{|G|} \sum_{g \in G} \tilde{g} \tilde{H}_{\text{MF}}^{(c)} \tilde{g}^{-1}. \tag{25}$$

Given that the reduced DM is constructed from the eigenstates of the Hamiltonian, it should naturally respect the corresponding symmetries inherited from the symmetrized Hamiltonian $\tilde{H}_{\text{MF,Symm}}^{(c)}$.

### 4.1.3 Comparison with Cluster Variation Method

CVM is a variational method that minimizes the free energy,

$$F[\rho] = \text{Tr}(\rho H) + k_B T \text{Tr}[\rho \ln(\rho)], \tag{26}$$

with respect to the density matrix $\rho$. In Eq.(26), $\rho$ is the many-body DM for the entire system, $k_B$ is the Boltzmann constant, and $T$ is the temperature. For a Hamiltonian consisting of terms with finite-range interactions, it is sufficient to know up to $n$-point reduced DM,

$$\rho^{(n)}(\{i_1, i_2, \ldots, i_n\}) = \text{Tr}_{\mathcal{I} \setminus \{i_1, i_2, \ldots, i_n\}} \rho, \tag{27}$$

to evaluate the energetic contribution $\text{Tr}(\rho H)$ to the free energy, where the set $\{i_1, i_2, \ldots, i_n\}$ supports of the terms in the Hamiltonian. For example, in Heisenberg models with exchange interactions and Zeeman couplings, only $\rho^{(n=2)}(i_1, i_2)$ is required, where the distance $|i_1 - i_2|$ is bounded by the interaction range of the Hamiltonian.

The key idea of CVM is to use cluster entropies to evaluate the entropic contribution $\text{Tr}[\rho \ln(\rho)]$ to the free energy. The cluster entropies rely only on the reduced DMs supporting the clusters and are expected to decay rapidly when the linear size of the clusters exceeds the correlation length of the system [32]. The cluster approximation involves selecting a truncation in cluster size. A particular choice of truncation for models with Heisenberg exchange interactions is at the second order, i.e., $n = 2$, which is also known as the pair approximation [29–31]. In this approach, the variational equations, along with the consistency conditions of the reduced DM, lead to the effective mean-field Hamiltonians of conventional MFTs [30, 31] and the separability $\rho^{(n=2)}(i, j) = \rho^{(n=1)}(i) \otimes \rho^{(n=1)}(j)$ for $|i - j|$ exceeds the interaction range [29].

The separability in CVM arises as a consequence of the absence of long-range interactions and the truncation at $n = 2$. However, quantum fluctuations can manifest as entanglements over longer ranges, inherently determined by the nature of quantum states. In this sense, DMMFT improves upon CVM with pair approximation by considering the quantum entanglements over the range of the extended cluster and determining the reduced DM $\rho_c$ self-consistently, up to $n = |c|$.

Another question worth mentioning is whether the mean-field equations derived in Sec. 2 can be derived from the perspective of CVM. As demonstrated in Ref.29, conventional mean-field theories [30, 31] can be derived as special cases of CVM with pair approximation and constant-coupling approximation. Different choices of consistency conditions can lead to different conventional MFTs [30, 31]. In this article, we have derived the mean-field equations for DMMFT by assuming the separability for distant clusters. However, whether separability can emerge as a natural consequence of alternative approximations beyond pair approximation, as demonstrated for conventional MFTs in CVM in Ref. 29), remains an open question. We do not have an answer to this question and leave it for future research.

### 4.1.4 Comparison with Linked-Cluster Expansion

LCE is another cluster-based method used for analyzing quantum spin systems [33–37, 39, 40]. Beyond conventional MFTs, LCE accounts for corrections from quantum and thermal fluctuations represented as a series expansion in $x^n$, where $x = H_I/(k_B T)$ and $H_I$ is part of the Hamiltonian describing the fluctuations [33–35]. Considering the series as a perturbative expansion ordered by the power $n$, $x$ can exceed the radius of convergence of the series at sufficiently low temperatures, which poses a challenge for convergence in LCE.

This obstacle is better addressed by numerical LCE, where physical observables are calculated still in the same basis as in LCE but not perturbatively in temperature [36, 37]. Then, it is the correlation length, instead of the temperature, that determines the convergence of the series. Consequently, numerical LCE can converge at a lower temperature than perturbative LCE. However, if long-range order develops at low temperature, and the correlation length becomes larger than the cut-off scale of the cluster sizes, the convergence is no longer guaranteed [38]. Resummation algorithms, such as Wynn's algorithm, Brezinski's algorithm [80], and Euler's transformation [81], have been developed to accelerate the convergence of numerical LCE [38].

In DMMFT, the cluster partition is fixed at the beginning, eliminating the need for summations over various configurations of the clusters. Although the size of the clusters in DMMFT and numerical LCE both set the cut-off scale for the correlated fluctuations, DMMFT does not suffer from convergence problems even if long-range order develops at low temperatures because no explicit summations are required. Moreover, practical applications of numerical LCE often involve handling a large number of large clusters of various sizes (often solved by ED), which can be computationally demanding. In this regard, DMMFT can outperform LCE in computational efficiency, particularly at low temperatures.

### 4.2 Systems at Finite Temperatures

The extension of DMMFT for systems at finite temperatures is straightforward. In Eq.(9), we use the ground state of the effective Hamiltonian over the extended cluster to compute the reduced DM. At finite temperatures, assuming local thermal equilibrium, we can average the reduced DMs of the eigenstates weighted by the Boltzmann distribution. Specifically, let $\{E_\alpha, |\tilde{\phi}_\alpha\rangle\}$ be the eigensystems of $\tilde{H}_{\mathrm{MF}}^{(c)}$. Then, the thermally averaged reduced DM is given by

$$[\rho_c^{\mathrm{MF}}]_T = \sum_\alpha w_\alpha \rho_c^\alpha,$$

$$w_\alpha = \frac{\exp[-E_\alpha/(k_B T)]}{\sum_\beta \exp[-E_\beta/(k_B T)]}, \tag{28}$$

$$\rho_c^\alpha = \mathrm{Tr}_{\tilde{c}} |\tilde{\phi}_\alpha\rangle\langle\tilde{\phi}_\alpha|,$$

where the square bracket with a subscript $T$ indicates the thermal average. As discussed in Sec. 2.3, when $D_{\mathcal{H}_{\mathcal{C}_c'}} = 1$, DMMFT reduces to the conventional MFTs at zero temperature. It is

straightforward to verify that since $[\rho_c^{\mathrm{MF}}]_T \sim \exp[-\tilde{H}_{\mathrm{MF}}^{(c)}/(k_B T)]$ when $D_{\mathcal{H}_{C_c'}} = 1$, the extension of DMMFT according Eq.(28) also aligns with the conventional MFTs at finite temperatures.

However, it is worth noting that the thermal averaging in Eq.(28) does not fully capture all thermal fluctuations. Specifically, at low temperatures, it is often the gapless long wave length modes (Goldstone modes) that dominate the thermal fluctuations. However, due to the mean-field approximation inherent in DMMFT, these thermal fluctuations are frozen by the assumption of translational symmetry. Therefore, similar to all other mean-field methods, DMMFT tends to underestimate the impact of thermal fluctuations at finite temperatures.

## 4.3 Systems with Disorders

Disorders are inherent in experiments and have the potential to significantly alter the nature of delicate quantum phases, even in minute quantities, especially within frustrated systems. Hence, integrating disorders into the DMMFT framework is crucial.

In the presence of the disorders, the Hamiltonian of the extended cluster undergoes direct modification, denoted as

$$H[\mathcal{O}_{\bar{c}}] \rightarrow H^{(\delta)}[\mathcal{O}_{\bar{c}}], \tag{29}$$

where the superscript $\delta$ labels the types of the disorders. However, the process of averaging over disorders is more intricate than the thermal average discussed in Sec. 4.2.

Consider a straightforward scenario involving a single magnetic vacancy within the cluster $c$, where there can either be no vacancies or one vacancy on a site $i \in c$. In this case, we denote $\delta \in \Delta = \{0\} \cup c$, where $\delta = 0$ signifies no vacancies in the cluster $c$. The probability measure of the disorder configurations is $P_\Delta(\delta)\mathrm{d}\delta$. Analogous to Eq.(28), it is intuitive to define an average over the disorders as

$$[\rho_c^{\mathrm{MF}}]_{\Delta,\mathrm{ann}} = \int_\Delta \rho_c^{(\delta)} P_\Delta(\delta)\mathrm{d}\delta,$$
$$\rho_c^{(\delta)} = \mathrm{Tr}_{\bar{c}} |\tilde{\phi}_{(\delta)}\rangle\langle\tilde{\phi}_{(\delta)}|, \tag{30}$$

where $\tilde{\phi}_{(\delta)}$ is the ground state of the effective Hamiltonian $\tilde{H}_{\mathrm{MF}}^{(c),(\delta)}$, and the square bracket with a subscript $\Delta$ indicates the average over the disorder configurations. The physical meaning of $[\rho_c^{\mathrm{MF}}]_{\Delta,\mathrm{ann}}$ so-defined requires further clarification. When $[\rho_c^{\mathrm{MF}}]_{\Delta,\mathrm{ann}}$ is used to set the effective environment $\mathcal{C}_c' = \{c'\}$ for the cluster $c$, each $c'$ is already averaged over various disorder configurations according to Eq.(30). In other words, the disorders are *annealed*, hence the subscript "ann" in Eq.(30).

In experiments, magnetic vacancies often exhibit behavior more akin to static disorders than fluctuating disorders, particularly when the vacancies do not thermally migrate. Such disorders are often referred to as *quenched* disorders, in contrast to the annealed disorders. In the case of quenched disorders, we consider all possible disorder configurations over the extended cluster $\bar{c}$. Let $\bar{c}$ consist of $\nu$ simple clusters, including the focused cluster $c$ and $\nu - 1$ clusters in the environment. Assuming the simple clusters are identical, all possible disorder configurations of the quenched disorders form a set $\boldsymbol{\Delta} = \Delta^\nu = \{(\delta_1, \delta_2, \ldots, \delta_\nu)\}$. We derive coupled mean-field equations for $\{\rho_c^{(\delta)}\}_\Delta$. Once $\{\rho_c^{(\delta)}\}_\Delta$ is solved, the expectation values of local observables averaged over the quenched disorders are calculated as

$$[\langle O_i^\alpha\rangle]_{\boldsymbol{\Delta},\mathrm{quen}} = \int_{\boldsymbol{\Delta}} \mathrm{Tr}_c \left[O_i^\alpha \rho_c^{(\delta)}\right] P_{\boldsymbol{\Delta}}(\boldsymbol{\delta})(\mathrm{d}^\nu\boldsymbol{\delta}), \tag{31}$$

where $P_{\boldsymbol{\Delta}}(\boldsymbol{\delta})(\mathrm{d}^\nu\boldsymbol{\delta})$ is the probability measure over the configuration space of the quenched disorders $\boldsymbol{\Delta}$. In the special case where each vacancy is independent and subjected to an identical distribution $P_\Delta(\delta)\mathrm{d}\delta$, $P_{\boldsymbol{\Delta}}(\boldsymbol{\delta})(\mathrm{d}^\nu\boldsymbol{\delta}) = \prod_{n=1}^\nu P_\Delta(\delta_n)\mathrm{d}\delta_n$.

In real systems, the effects of disorders can be more complicated. For instance, there can be clusters of vacancies due to lower formation free energy, and the probability measure of the vacancies may not take a simple product form. Moreover, disorders can manifest as inhomogeneity in the samples, especially when scale of the probe is small, and the effects of disorders are not self-averaged in the experiments. Despite the complications of disorders arising in real systems, DMMFT can, in principle, be adapted to include these disorder effects accordingly.

# 5 Conclusion

In this study, we introduced a generalized mean-field approach, DMMFT, specifically designed to capture quantum fluctuations beyond conventional MFTs. DMMFT combines the strengths of DMRG and DMFT, and stands out as an efficient and unbiased method applicable to fermions, bosons, and spins, even in the presence of frustrations. A notable feature of DMMFT is its capability to discern topological phases, enabled by its utilization of the reduced DM, as demonstrated with the example of the AKLT model in Sec. 3.1. Furthermore, DMMFT can achieve precision comparable to DMRG, even with a small cluster, as demonstrated with the example of the AFHTL in Sec. 3.2. Additionally, DMMFT seamlessly extends to systems at finite temperatures and incorporates disorders. In conclusion, DMMFT provides an effective tool for exploring phases exhibiting unconventional quantum orders and is particularly beneficial for investigating frustrated spin systems in high spatial dimensions.

# Acknowledgements

JYZ thanks to Yi Li, Collin Broholm, Oleg Tchernyshyov, and Tong Chen for their encouragement and discussions. JYZ also thanks Salvia Felixsen Skogen for the hospitality at the Domini Katzen Institute during his visit to Columbia University.

**Funding information**     JYZ acknowledges support from the NSF CAREER Grant DMR-1848349 and from the Johns Hopkins University Theoretical Interdisciplinary Physics and Astronomy Center. JYZ also acknowledges the Institute for Quantum Matter, an Energy Frontier Research Center funded by the U.S. Department of Energy, Office of Science, Office of Basic Energy Sciences, under Award DESC0019331, for support during the initial stage of this work on the triangular lattice material $K_2Co(SeO_3)_2$. ZC acknowledges the support by the Columbia Center for Computational Electrochemistry.

# A   Iterative Algorithm for Mean-Field Equations

In Section 2.2, we established a closed set of coupled mean-field equations for DMMFT, delineated by Eqs. (9) to (14). In this section, we present an iterative algorithm, detailed in Table 1, to achieve self-consistent numerical solutions.

There are several important considerations to keep in mind. In Step 0, although a stable self-consistent solution should be independent of the initial values, providing an initial guess close to the self-consistent solution often aids in convergence. An empirical approach is to start with a trivial $\rho_c$ that can be constructed from solutions of conventional MFTs. In Step 0',

| Step number | Step description |
|---|---|
| 0 | Initialize a set of trial reduced DMs $\{\rho_c\}_{c\in\mathcal{C}}$. |
| For each cluster $c$: | |
| 1 | Diagonalize $\rho_c$, and construct $\Pi_c$ according to Eq.(13). |
| 2 | Use Eq.(14) to construct $\Pi_{\mathcal{C}'_c}$. |
| 3 | Use Eq.(10) to construct $\Pi$. |
| 4 | Use Eq.(11) to construct $\tilde{H}^{(c)}_{\mathrm{MF}}$. |
| 5 | Diagonalize $\tilde{H}^{(c)}_{\mathrm{MF}}$ and select the target state $|\tilde{\phi}_c\rangle$. |
| 6 | Calculate new $\rho_c$ according to Eq.(9). |
| 0' | Use new $\{\rho_c\}_{c\in\mathcal{C}}$ to update the trial reduced DMs, and repeat steps 1 – 6 until convergence. |

Table 1: DMMFT implementation steps.

a linear interpolation of the reduced DM,

$$\rho_c^{(n+1)} = (1-\alpha)\rho_c^{(n)} + \alpha\rho_c^{(n,\mathrm{new})}, \tag{32}$$

can be employed for updating $\rho_c$. Here, $\rho_c^{(n)}$ represents the reduced DM used in Step 0 at the $n$-th iteration, and $\rho_c^{(n,\mathrm{new})}$ is the new reduced DM obtained in Step 0', The learning rate $\alpha$ varies in the range $(0, 1)$. Larger values of $\alpha$ lead to a more rapid update, while smaller values tend to stabilize the iterative process.

Finally, the hyperparameter $n_c$ used in Eq.(13) in Step 1 can be compared to $\tilde{D}^E_c = \lceil \exp(SE_c) \rceil$, where the entanglement entropy is calculated according to Eq.(8). It is important to note that setting $n_c = 1$ should recover the results of conventional MFTs.

## B   Pseudocodes for Heisenberg Model

In this Appendix section, we describe the implementation of DMMFT for the AFHTL as discussed in Sec. 3.2.

We partition the triangular lattice into clusters, each comprising three lattice sites, as shown in Fig. 4. The hexagons represent the Wigner-Seitz cells of the three-lattice clusters, which are compatible with the symmetries of the underlying triangular lattice. The central cluster $c$ is highlighted in orange. We choose an extended cluster $\bar{c}$ comprising of four three-lattice clusters $\{c; c'_1, c'_2, c'_3\}$ with periodic boundary conditions. Fig. 4 depicts the extended cluster in a periodically extended scheme, where clusters in $\mathcal{C}'_c$ are duplicated under periodic translations $\mathbf{a}_i$. Bonds depicted with solid lines indicate intra-cluster couplings contained in $H_c$, while those with dashed lines indicate inter-cluster couplings contained in $H_{c,\mathcal{C}'_c}$. For visual clarity, Fig. 4 abbreviates some coupling bonds.

Now, we proceed to describe the pseudocode for DMMFT.

First, we define the local degrees of freedom. For the AFHTL, each site contains a quantum spin of $S = \frac{1}{2}$. We define the spin operators $S_i^\alpha, \alpha = x, y, z$ and the identity operator for each site as follows.

```
1  '# Code Block 1: define the physical site: S=1/2 #'
2  % length of the spin
3  slen = 1.0/2.0;
4  % dimension of local Hilbert space
5  dimHi = 2;
6  % define local operators
```

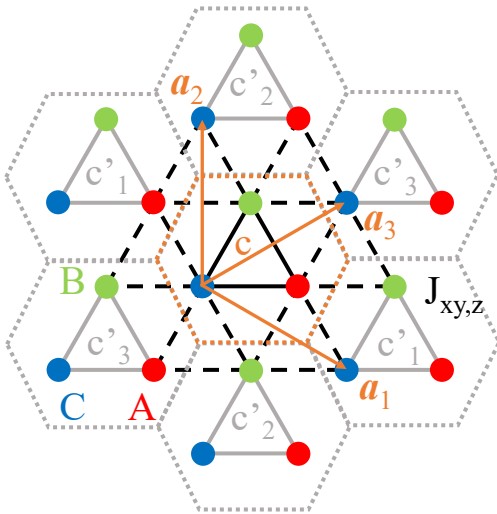

Figure 4: Extended cluster with the periodic boundary condition. The hexagons represent the Wigner-Seitz cells of the three-lattice clusters compatible with the symmetries of the underlying triangular lattice. The central cluster $c$ is highlighted in orange. Three neighboring clusters $\mathcal{C}'_c = \{c'_1, c'_2, c'_3\}$ are duplicated under periodic translations $\mathbf{a}_i$. Bonds depicted with solid lines indicate intra-cluster couplings contained in $H_c$, while dashed lines indicate inter-cluster couplings contained in $H_{c,\mathcal{C}'_c}$. For visual clarity, not all coupling bonds are shown.

```
7 idiop = eye(dimHi);
8 sxop  = [0.0,   1.0;   1.0,   0.0]*slen;
9 syop  = [0.0, -1.0j;   1.0j,  0.0]*slen;
10 szop = [1.0,   0.0;   0.0,  -1.0]*slen;
```

Next, we define the local operators for the clusters shown in Fig. 4. These local operators for cluster $c$ are constructed from the site operators using the tensor product, and then represented in matrix form using the function kron().

```
1 '# Code Block 2: define local operators for cluster c #'
2 % number of sites in c
3 nsite = 3;
4 % dimension of the cluster Hilbert space
5 dimHc = dimHi^nsite;
6
7 idcop = eye(dimHc);
8 opclist = {idcop};
9
10 for isite in c:
11   % construct siaop as tensor products of idiop's saop
12   % for example: s1xop = kron(sxop,kron(idiop,idiop))
13   opclist.append(siaop);
14 end for % isite
```

Using the local operators, we construct the Hamiltonian $H_c$, which includes intra-cluster exchange interactions and Zeeman couplings, according to their definitions as follows.

```
1 '# Code Block 3: construct Hamiltonian of intracluster couplings #'
2 Hamc = zeros(dimHc);
3 % add exchange interactions
4 for ibond in c:
5   i,j = vertex(ibond);
```

```
6    Hamc = Hamc + sum_a Ja*siaop*sjaop;
7  end for % ibond
8  % add Zeeman couplings
9  for isite in c:
10    Hamc = Hamc - sum_a ha*siaop;
11 end for % ibond
```

Then, we define the operators for the extended cluster $\bar{c}$.

```
1  '# Code Block 4: define operators for the extended cluster cbar #'
2  % number of clusters in cbar
3  ncbar = 4;
4  % dimension of the cluster Hilbert space
5  dimHcbar = dimHc*dimHcp^(ncbar-1);  % cf. Eq.(10)
6
7  idcbarop = eye(dimHcbar);
8  opcbarlist = {idcbarop};
9
10 % construct spin operators
11 for isite in cbar:
12   % construct siabarop as tensor products of idcop's siaop
13   % for example: s1xbarop = kron(s1xop,kron(idcop,kron(idcop,idcop)))
14   opcbarlist.append(siabarop);
15 end for % isite
16
17 % construct Hamiltonian operator with intra- and inter-cluster terms
18 Hamcbar = sum_c Hamc + sum_{c,cp} Hamccp;
```

Finally, we solve the mean-field equations self-consistently. We use the iterative method to solve the mean-field equations, following the steps described in Append. A.

```
1  '# Code Block 5:  Iterative solver #'
2  '> step 0: initialization <'
3  rhoc = reduced DM from classical spin configurations
4
5  % configure iteration process
6  % learning rate
7  alearn = 0.6;
8  % cut-off dimension, n_c
9  dimHcp = 4;
10
11 % start iterations
12 while (not converge):
13   '> step 1: construct Projc [Eq.(13)] <'
14   evec_rhoc, eval_rhoc = eig(rhoc);
15   Projc = zeros(dimHc,nc);
16   for istate in largest n_c evals of rhoc:
17     Projc.append(evec_rhoc(istate));
18   end for % istate
19
20   '>  step 2: construct ProjCp [Eq.(14)] <'
21   for ic in C':
22     ProjCp = kron(ProjCp,Projc)
23   end for % ic
24   '>  step 3: construct Proj [Eq.(10)] <'
25   Proj = kron(idcop,ProjCp);
26
27   '>  step 4: construct effective Hamiltonian [Eq.(11)] <'
28   Ham_MFeff = Proj'*Hamcbar*Proj;
29
30   '>  step 5: find ground state of Ham_MFeff <'
31   evec_Ham, eval_Ham = eig(Ham_MFeff);
32   phicbar = state in evec_Ham with lowest energy;
33   rhocbar = phicbar*phicbar';
```

```
34
35   '>  step 6: calculate new rhoc [Eq.(9)] <'
36   rhoc_new = partially tracing rhocbar over {c'};
37
38   '>  step 0: update rhoc [Eq.(29)] <'
39   rhoc = (1-alearn)*rhoc + alearn*rhoc_new;
40   check convergence;
41
42 end while   % iteration
```

The pseudocode provided above serves only to demonstrate the step-by-step implementation of DMMFT. Various optimizations can enhance memory and time efficiency. For example, one may use the Lanczos algorithm in `Code Block 5`, `line 14` and `line 31`, since not the full spectra of $\rho_c$ and $\tilde{H}_{\mathrm{MF}}^{(c)}$ are used.

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
