# Peer review of "Density-Matrix Mean-Field Theory"

_SciPost Physics_

## Round 1 · Referee Report · Anonymous (Referee 2) · 2024-3-3

Report

The manuscript makes an interesting proposal, namely a new "Density-Matrix Mean-Field Theory" (DMMFT). However, whether or not this merits publication in SciPost Physics (see https://scipost.org/SciPostPhys/about#criteria for the Acceptance criteria) depends on the performance of the method. In this respect, chapter 3 of the manuscript is essential, but here I am not convinced [yet].

The application in section 3.1 to the Affleck-Kennedy-Lieb-Tasaki Model, i.e., the spin-1 bilinear biquadratic chain Eq. (18) with $\beta = \frac13$ is fairly simple and thus not very deep. Nevertheless, the statement "The AKLT model is an exactly solvable system, showcasing non-trivial topological order. Notably, it features spin-1/2 (fractional to spin-1) edge states and exhibits 4-fold degeneracy of the ground states for an open chain." would need to be substantiated with references. In fact, it would probably be possible to also investigate the case $\beta \ne \frac13$, but in this case there are further (numerical) investigations that would also need to be cited.

This renders section 3.2 about the antiferromagnetic spin-1/2 Heisenberg model on the triangular lattice central. Here, the real benchmark would not be conventional mean-field theory, but numerical results that are available in the literature. In fact, I would consider a quantitative comparison more important than the generic comparison with alternative methods that is presented in section 4.1. I also note that in the related list of references [31-47] that are cited in the Introduction, not all relevant ones are mentioned, and not all cited are indeed relevant (e.g., some of these references concern only the zero-field case that is not really relevant here, and some concern the $J_1$-$J_2$ model, where only the special case $J_2=0$ is relevant here). Previous publications on the XXZ model that might be relevant include A. Honecker, J. Schulenburg, and J. Richter, J. Phys.: Condens. Matter 16, S749 (2004) and D. Sellmann, X.-F. Zhang, and S. Eggert, Phys. Rev. B 91, 081104(R) (2015) as well as J. Richter, J. Schulenburg, A. Honecker, Lect. Notes Phys. 645, 85 (2004) (specifically Sec. 2.6.2 loc. cit.) and T. Sakai and H. Nakano, Phys. Rev. B 83, 100405(R) (2011) for the Heisenberg model. There may be more recent relevant references, but there definitely are older ones; the latter are cited in the publications mentioned here.

I note in passing that there is spurious lowercasing in the titles of the references. For example, the "XXZ" in the title of Ref. [42] that is central to the present work is misprinted and names such as "Néel" in the title of Ref. [28] should not be lowercased either.

Without further discussion of the quantitative accuracy of DMMFT, I believe that the present work would be more suitable for SciPost Physics Core, see https://scipost.org/SciPostPhysCore/about#criteria.
  • validity: high
  • significance: good
  • originality: high
  • clarity: high
  • formatting: excellent
  • grammar: excellent

Author:  Junyi Zhang  on 2024-04-25  [id 4449]

(in reply to Report 2 on 2024-03-03)
Category:
remark
answer to question
objection
correction

We thank the referee for being interested in our work and one’s consideration of the method's performance. In response to this concern, we have provided a benchmark comparison (Fig. 3) with the state-of-the-art DMRG method in this revised manuscript. The comparison demonstrates good alignment between our DMMFT results and the DMRG results, outperforming the conventional mean-field theory (CMFT).
Moreover, it is noteworthy to emphasize the efficiency of our DMMFT compared to DMRG. While DMRG calculations for a cylindrical patch of $6\times 30$ containing 180 sites require over 120 CPU hours, DMMFT achieves comparable precision within a few CPU minutes using a small three-lattice cluster. This efficiency advantage becomes even more significant for systems in higher dimensions. We hope this benchmark example effectively illustrates the performance capabilities of our method.

We appreciate the referee's feedback and would like to address concerns regarding the application of the Affleck-Kennedy-Lieb-Tasaki (AKLT) model in section 3.1. While we acknowledge that the AKLT model may appear simple at first glance, we believe it offers valuable insights that merit consideration.
Firstly, the simplicity of the AKLT model enables both exact analytical and numerical solutions, making it an ideal benchmark example. However, its simplicity does not compromise its generality, as it exhibits topological properties that are robust against perturbations, not limited to the special integrable point of β=1/3. Historically, the AKLT model was proposed to confirm Haldane's conjecture on spin-1 chains, demonstrating its significance in understanding quantum phases.
Our choice of the AKLT model as a benchmark example serves to highlight DMMFT's capability in detecting topological phases through quantum entanglement, which surpasses conventional MFTs. Moreover, in contrast to conventional MFTs, which fail to detect orders in ground states with vanishing conventional order parameters, DMMFT utilizes the reduced density matrix (always well-defined in the many-body Hilbert space) as a general local observable to identify topologically ordered ground states based on quantum entanglement. This provides deeper insights into quantum ordered phases beyond conventional MFTs, which may be further extended to scenarios like RVB spin liquid phases in higher dimensions.
Furthermore, we have included a quantitative comparison with the state-of-the-art DMRG method for the AFHTL. The results demonstrate excellent agreement between DMMFT and DMRG, highlighting the method's effectiveness in capturing the complexities of frustrated systems in two dimensions.
In summary, while the AKLT model may seem simple, its inclusion in our study serves to showcase the capabilities of DMMFT in detecting quantum order beyond the limitations of conventional MFTs. Additionally, our quantitative comparison with DMRG for the AFHTL in 2D further validates the efficacy of DMMFT in capturing the intricacies of frustrated systems.

We appreciate the referee's request for substantiating the statement regarding the AKLT model's properties with relevant references. The ground state wave functions and properties of the AKLT model have been rigorously established in the seminal work by Affleck, Kennedy, Lieb, and Tasaki in Commun. Math. Phys. 115, 477-528 (1988) [Ref. 43 of the revised manuscript], after whom the model is now named.
Particularly, Eq. 2.7 in that paper gives an explicit expression of the ground state wave functions showcasing the model's non-trivial topological order. Additionally, the ground state degeneracy of the AKLT model has been explicitly discussed in Remark 1 on page 484 of the same reference. This groundbreaking work by Affleck et al. stands as the foundational study establishing the exact quantum ground state of many-body systems and serves as the cornerstone for understanding the AKLT model's unique properties, including its fractional spin-1/2 edge states and the 4-fold degeneracy of the ground states for an open chain.

We appreciate the referee's suggestion regarding the investigation of cases where β≠1/3 in the context of the AKLT model. While we acknowledge the potential interest in exploring such cases, we believe that for the purpose of demonstrating the effectiveness of DMMFT, focusing on the β=1/3 case suffices for several reasons.
Physically, the AKLT model serves as a prototype for studying topological phases due to its integrability. However, when β≠1/3, the model is no longer integrable. It is widely accepted that for 0<β<1/3, the models remain within the same universality class as the AKLT model, with the Heisenberg model (β=0) being on the boundary of this universality class, as stated in Ref. 47 and 55 in the revised manuscript.
From our perspective, the application of DMMFT to the AKLT model does not rely on its integrability. Therefore, our conclusion regarding DMMFT's ability to detect topological phases remains valid for the class of topological phases for general 0<β<1/3.
Furthermore, considering the AKLT model as a realization of a quantum antiferromagnet with vanishing Néel order, our work, along with the discussion of the anisotropic ferromagnetic Heisenberg model in Section 3.2, sufficiently demonstrates DMMFT's capability to correctly characterize quantum magnetic phases, irrespective of the semiclassical order parameter.
Given these considerations, we believe that additional calculations for the 1D AKLT model with general β≠0, 1/3 may not provide significant additional insights for our current purposes. DMMFT's advantages are expected to be more pronounced in higher spatial dimensions, while β-deformed AKLT models can be efficiently studied with DMRG in 1D.

We agree with the referee that quantitative comparisons can serve as more concrete support for our new method. In response to this suggestion, we have conducted additional DMRG calculations for the AFHTL with an anisotropic parameter A=0.9. These calculations allow for a direct quantitative comparison with our DMMFT results, providing more concrete support for our method.
Upon reviewing the literature, we found that while various studies have investigated AFHTL and reported phase diagrams, precise magnetization curves for the specific case of A=0.9 were not readily available. To address this, we conducted our own DMRG calculation for A=0.9, allowing for a direct comparison with our DMMFT results.
We also consulted relevant literature suggested by the referee, such as the work by Yamamoto, Marmorini, and Danshita [J. Phys. Soc. Jpn. 85, 024706 (2016), Ref. 53], which utilized exact diagonalization, cluster mean-field theory with scaling, and the coupled cluster method. Although this provided valuable insights, the lack of supplemental data hindered a more quantitative comparison.
In our revised manuscript, we present the magnetization curve obtained from our DMRG calculation alongside our DMMFT results. This comparison reveals excellent agreement between the two methods, particularly at low fields, demonstrating that DMMFT accurately captures corrections from quantum fluctuation effects compared to CMFT.
Additionally, our DMRG calculation allows us to estimate the critical field $h_{c2}$, which is found to be larger than that predicted by conventional MFT but smaller than our DMMFT result. We acknowledge that finite-size effects in the transverse dimension may have influenced the accuracy of our DMRG result for $h_{c2}$, and our choice of cluster size in DMMFT may have led to an overestimation of the critical field. Nonetheless, deeper exploration into the "V"-shaped phase reveals better agreement between DMRG and DMMFT compared to CMFT.
These quantitative comparisons, presented in Fig. 3 of the revised manuscript, provide strong evidence supporting the effectiveness of DMMFT in capturing the quantum behavior of AFHTL. We have incorporated these findings into our revised manuscript and adjusted the discussions accordingly to reflect the comparisons with DMRG results.

We appreciate the referee's feedback regarding the relevance of the cited references. While we acknowledge that we only discussed in detail the simple AFHTL with nearest-neighbor exchange couplings, we believe it is important to recognize the broader context in which our research resides.
The $J_1$-$J_2$ Heisenberg model and other variants, although not discussed in detail in this paper, are indeed relevant to the broader understanding of frustrated magnetism. Therefore, we included references to these relevant works in our list of citations to provide readers with a more comprehensive perspective on both historical and cutting-edge research in the field. While our immediate focus may be on the AFHTL, acknowledging the broader landscape of related models enriches the context of our research and may inspire further exploration into the diverse phenomena of quantum magnetism.

We appreciate the referee's suggestion to include additional references related to the XXZ and Heisenberg models. The references provided by the referee indeed offer valuable insights into the XXZ and Heisenberg models, which contribute to our understanding of frustrated magnetism and quantum phases. We thank the referee for highlighting these important references, and we have incorporated them into our revised manuscript accordingly.

We appreciate the referee's attention to detail regarding the formatting of citations. We have carefully reviewed the references and made the necessary corrections to ensure that all chemical compounds, people's names, and model names are formatted correctly, overriding the default compilation bib style. As a result of this review, the misprinted titles, such as “XXZ” and names like “Néel” in the titles have been corrected to their proper formatting.

We appreciate the referee's suggestion regarding the suitability of our work for SciPost Physics Core. In response to the referee’s feedback, we have made significant enhancements to our manuscript by supplementing DMRG calculations and conducting quantitative comparisons with DMMFT calculations. The results of our DMMFT method align well with those obtained from DMRG, demonstrating its accuracy and reliability. Furthermore, our method achieves comparable precision to DMRG while requiring significantly less computational resources, with a three-site cluster achieving results on par with DMRG. We hope these quantitative comparisons and the efficiency of our method provide better support for its publication in SciPost Physics.

---

## Round 1 · Referee Report · Anonymous (Referee 1) · 2024-3-3

Strengths

The manuscript under consideration proposes a novel formulation of the mean-field theory by accounting for short-range entanglement via the reduced density matrix of the subsystem. The idea makes sense and can be useful to high-dimensional ordered systems (higher than one-dimensional) with significant quantum fluctuations.

Weaknesses

The approach is not intuitive and a simple illustration of it on some simple toy model would help to understand it better.

Report

I can not comment on the numerical side of the proposal, but the two applications of the approach - the Haldane chain and the triangular XXZ spin-1/2 model in an external magnetic field - look quite promising. The authors find that (in the second example), the 1/3 magnetization plateau (the UUD phase) extends relative to the conventional cluster mean-field theory. This signals that DMMFT captures more quantum fluctuations than standard MFT.

Requested changes

Fig.2b also shows that M(h) behavior of MFT and DMMFT calculations is different below h_{c1}, the red DMMFT curve is quite nonlinear. This leads to the question: which of these is closer to the correct M(h) curve? It would be good for authors to comment on this and/or to compare it with the “exact” DMRG result if that exists.

  • validity: good
  • significance: good
  • originality: high
  • clarity: good
  • formatting: good
  • grammar: good

Author:  Junyi Zhang  on 2024-04-25  [id 4448]

(in reply to Report 1 on 2024-03-03)

We appreciate the referee's suggestion to compare our results with those obtained from DMRG calculations. To provide a more concrete comparison, we have performed DMRG calculations and included the results in this revised manuscript. Particularly, Fig.3 of the revised manuscript shows great agreement between the results DMMFT and the results of DMRG.
To compare with results reported in previous literatures with DMRG or alternative methods, we searched previous studies of Heisenberg model on triangular lattice in external magnetic field. While various studies have investigated the phase diagrams in the literature, we did not find magnetization curve reported specifically for the anisotropic parameter A=0.9. One relevant study we found is by Yamamoto, Marmorini and Danshita, J. Phys. Soc. Jpn. 85, 024706 (2016), Ref. 53 in the revised manuscript, where they employed exact diagonalization, cluster mean-field theory plus scaling, and coupled cluster methods. Particularly, in Fig. 4(a) in Ref. 54, for small clusters, they presented magnetization curves for small clusters, which exhibit linearity, consistent with our calculations with conventional MFT. Furthermore, their extrapolated magnetization curve for an infinite system shows discernible nonlinearity, which also aligns with our DMMFT results. Unfortunately, there were no supplemental data provided allowing more quantitative comparison. Similar reports may also be found in Ref. 47-49, although they focus on the isotropic case.
To facilitate a more quantitative comparison, we have performed DMRG calculations and included results in this revision. Specifically, we simulated the AFHTL system on a lattice of $6 \times 30$ sites in a cylindrical geometry with periodic boundary conditions along the shorter edge, using the ITensors package. The results obtained from DMMFT and DMRG have been overlaid in Fig. 3 of the revised manuscript. Upon comparison with the DMRG results, we observe that at low fields, the magnetization curve exhibits nonlinear behavior, in good agreement with our DMMFT results. This supports the conclusion that DMMFT can more accurately capture corrections from quantum fluctuation effects compared to conventional mean-field theory (CMFT).
Furthermore, the DMRG calculations also provides an estimation of $h_{c2}$. Our DMRG results shows $h^*_{c2}=2.06$, which is larger than the value of CMFT $h_{c2}=2.0$ but smaller that from DMMFT $h’_{c2}=2.2$. While our DMRG calculation unequivocally demonstrates $h^*_{c2} > h_{c2}$, we believe this value may still underestimate the true critical field due to finite size effects in the transverse dimension. This is consistent with trends observed in phase diagrams from previous studies (Ref. 47-49, 52,53), particularly as illustrated in Fig. 1 of Ref. 48. On the other hand, the critical field obtained from our DMMFT results may have been overestimated, as we have chosen $n_c=4$, which is larger than the optimal cut-off dimension estimated from the entanglement entropy. Moreover, deeper into the “V”-shaped phase, DMRG results show better agreement with DMMFT than CMFT.
In summary, we have included our DMRG calculations in this revised manuscript for more quantitative comparisons, and we have modified our discussions to incorporate comparisons with DMRG results accordingly.

We regret any confusion caused by the abstract presentation of our theory in Section 2. While we aimed to provide a balance between generality and accessibility, we understand the importance of offering intuitive examples for better comprehension.
To address this concern, we have made several enhancements in this revised manuscript. In addition to Section 3 and Appendix A, we have added another section, Appendix B, which provides pseudocodes for implementing DMMFT specifically for the AFHTL example discussed in Section 3.2. This addition aims to offer practical guidance for understanding and implementing DMMFT.
Furthermore, in Appendix A, we have included a flowchart outlining the general iterative algorithm for solving DMMFT equations. Each step in the algorithm is accompanied by explicit references to the corresponding equations, providing a clear roadmap for readers to follow.
We hope that these additions will improve the clarity and accessibility of our methodology, enabling readers to better grasp the concepts and facilitate their implementation. Furthermore, in line with the spirit of Scipost series of journals, which promotes open access, our source codes are available up on request to the authors.

---

## Round 1 · Referee Report · Anonymous (Referee 3) · 2024-3-4

Report

The manuscript by Zhang and Cheng discusses a novel mean-field theory for interacting quantum systems, the so-called DMMT approach. The writing presents the formalism and discusses its relation to comparable schemes, such as DMFT, DMET and DMRG.
The text ist well written, the method becomes accessible and the generally revealed physics appears sound. Main idea appears to separate the full quantum-lattice problem into the more manageable problem of individual clusters building up the lattice under consideration. That idea is heavily used on various levels of interacting lattice problems. In quantum theory, the linked cluster expansion (LCE) approach is e.g. well known for that picture. In classical statistical mechnics, the cluster variation method (CVM) takes on that role. However, both techniques are not mentioned in the present context.
Thus, aside from the comparisons to DMFT, DMET and DMRG, the authors should also connect their approach to LCE and CVM. In addition, the crucial role of cluster nestings (i.e. overlapping clusters) and the issue of translational invariance (e.g. being a key obstacle in the various flavors of cluster-DMFT) is only scarcely touched. This aspect should be discussed more clearly and concrete statements in that respect are needed.
If the authors take the aforementioned points of criticism into account in a revised version of their manuscript, support for publication may be granted.
  • validity: -
  • significance: -
  • originality: -
  • clarity: -
  • formatting: -
  • grammar: -

Author:  Junyi Zhang  on 2024-04-25  [id 4447]

(in reply to Report 3 on 2024-03-04)
Category:
remark
answer to question

We appreciate the referee’s perspective on connecting our approach to linked cluster expansion (LCE) and cluster variation method (CVM). We agree that discussing these alternative methods can provide valuable insights for readers. We aim to provide a clearer understanding of DMMFT in relation to other cluster methods such as LCE and CVM by highlighting their connections and distinctions. We have incorporated these discussions into our manuscript by adding two new subsections (sec. 4.1.3 and 4.1.4) devoted to CVM and LCE respectively to enhance its comprehensiveness and clarity. For convenience of the referee’s, we summarize our discussions of LCE and CVM compared to DMMFT as follows. 1) LCE is an expansion method in terms of clusters and requires summation over various configurations of the clusters that contribute to the quantum fluctuations. More precisely, in LCE, the expansions are taken with respect to an energy scale, often chosen as temperature, which obstacles the convergence at low temperature. Moreover, in numerical LCE, the temperature is treated non-perturbatively, but the obstacle of convergence still exists, being the correlation length compared to the cluster size instead of the temperature. In DMMFT, the partition of the lattice into clusters is fixed in the first step, therefore there are no summation processes, and instead one looks for self-consistent solutions to the mean field equations. In this sense, our method is closer to the “cavity” in the dynamical mean-field theory (DMFT) or the clusters in various cluster extensions of DMFT, and not affected by the convergence problem. Moreover, in DMMFT, the major approximation with the cut-off scale of the clusters are in quantum entanglement, therefore DMMFT does not suffer convergence problem in case of a long range order developing at low temperatures (as shown in our demonstrative example of Heisenberg model on triangular lattice at zero temperature). Furthermore, DMMFT is more computational efficient than (numerical) LCE. In LCE, the summation is taken over various cluster configurations, and each in practice is handled with ED, which can be very computational demanding for moderately large cluster size due to the rapidly increasing combinatorial counting of the cluster configurations. In DMMFT, the cluster are fixed and no summation is needed, even up to comparable cluster sizes, DMMFT is expected to perform more efficiently than LCE. 2) CVM is a variational method minimizing the free energy with respect to the density matrices. The conventional MFTs can often be regarded as CVM with a cut-off at two-particle reduced DM (so-called pair approximation). From the perspective of CVM, DMMFT improves over the conventional MFTs by considering the reduced DM up to $|c|$-particle reduced DM, i.e., of the size of the clusters, self-consistently.

We also thank the referee for highlighting the importance of discussing cluster nestings and translational invariance more clearly, and we appreciate the opportunity to address these technical aspects in greater detail. Regarding cluster nestings, we would clarify in short that our method is not affected by consistency conditions due to cluster nesting. The key to this observation is that the lattice is partitioned into non-overlapping clusters, and the intercluster couplings only appears through the local projectors of the form $Id_{c} \otimes \Pi_{\mathcal{C}’_c}$ without using the reduced density matrix over the entire extended cluster. However, more generally, the problem of constructing a reduced DM for a general collection of the clusters fall in the category of the so-called quantum marginal problems, which we briefly touched in the end of Sec.4.1, and it remains an open research question. Nevertheless, the separable Ansatz of direct products of local reduced DM cannot be too bad, since if we take $n_c=1$ it becomes a pure state that is exactly the self-consistent wave function of the conventional MFTs. The problem of translation invariance, or more generally the lattice symmetry, is a shared problem not only to DMFT, but to all cluster approach that is formulated in real space. In cluster extensions of DMFT, this problem arises as the approximate self-energy does not preserve the symmetry of the under lying lattice, and periodization needs to be imposed artificially. Subtle issues can arise in the context of DMFT. While in DMMFT, the symmetrization can be imposed more straightforwardly by symmetrizing the effective Hamiltonian directly as shown in Eq. 25 in the revised manuscript.

We appreciate the referee’s feedback and have carefully addressed the points of criticism in the revised version of our manuscript. We believe that the revisions have significantly improved the clarity and comprehensiveness of our work.

---

## Round 2 · Referee Report · Anonymous (Referee 6) · 2024-5-13

Report
With their revised version, the authors have significantly improved and expanded their manuscript. The revised manuscript thus makes a convincing case that "Density-Matrix Mean-Field Theory" (DMMFT) is a promising new tool for the analysis of frustrated spin systems and other challenging many-body problems.
Nevertheless, I stumbled across the use of "integrable" in their reply. In fact, in the AKLT model with β=1/3, one can write down the exact ground state in closed form. However, this does not mean that other properties such as excited states are also amenable to an exact treatment, i.e., that the model is integrable in the Bethe-ansatz sense. The bilinear-biquadratic spin-1/2 chain does have two integrable points, namely the Uimin-Lai-Sutherland point at β=1 and the Takhtajan-Babujian point at β=−1 (see, e.g., chapter 1.5.2 of H.-J. Mikeska and A.K. Kolezhuk, Lect. Notes Phys. 645, 1 (2004)), but the AKLT point β=1/3 does not fall into this class. In view of SciPost Physics also serving the community working on integrable systems, I recommend to get this notation straight. This implies that related terminology in chapters 3 needs to be revised, with several instances on page 8 and one on page 9.
There are a number of further small details that I noticed when rereading the revised manuscript and I list these among "Requested changes". Maybe some of these could have been addressed already during the previous round and I apologize for bothering the authors with these now. However, since this promises to be an excellent paper, it may nevertheless be a good idea to iron out also the last details.
Requested changes
1- Reserve terminology such as "integrable"or "exactly solvable" to models solvable in the Bethe-ansatz sense, i.e., use different terminology for the exact ground state of the AKLT model.
2- End of abstract: claims for novelty are always a bit delicate and such judgment is better left to the reader. Furthermore, the authors do not really need this. I thus suggest to shorten "a novel and efficient approach" to "an efficient approach".
3-Top of page 3: "classical" is misspelled ("calssical").
4- Third paragraph on page 3: something missing in "of the Ising model using [28]" (or spurious "using")?
5- First paragraph of section 3: attributing the string-order parameter to the fairly recent Ref. [60] is a bit strange. In fact, Ref. [60] refers back to Ref. [68] in this context. At the very least, I suggest to also cite Ref. [68] at this place.
6- Second paragraph of section 3: "of" and "Anderson's" looks redundant to me (I would skip the "'s").
7- Figure 1: Looking again at it, I am not sure if the meaning of the horizontal axis was actually explained. Likewise, I can see error bars for DMMFT in the left panel, but it is not clear where they come from (after all, there is no statistical error here, or is it?). Maybe the authors could add some brief comments to clarify.
8- 4 lines below Eq. (22): spurious ' in '"Y".
9- Footnote 1 is kind, but maybe not necessary in view the reports being public such that this is on record. However, this is definitely a point where the authors are free to decide what they prefer.
10- Figures seem mostly to be images. I noticed this on a printout where Fig. 2 came out in borderline quality. If vector graphics (PDF) versions could be provided throughout, this might ensure absence if resolution issues.
11- Page 13: "literature[s]" in "in the literature" should in my opinion not have a plural "s".
12- Bottom of page 13: stabilization of the UUD state by quantum fluctuations has a history that goes well back past Ref. [52], see for example A.V. Chubukov and D.I. Golosov, J. Phys.: Condens. Matter 3, 69 (1991).
13- Between Eqs. (26) and (27): I think an article ("a"?) is missing in "For Hamiltonian".
14- There are two "to to" on the two lines preceding Eq. (30).
15- Top of page 15: "there can be cluster" - something missing, either an article or a plural "s"?
16- Refs. [3,9,16,31,53]: An URL in addition to a DOI is not needed.
17- Refs. [24,25]: "Mott" respectively "Hubbard" should start with a capital letter.
18- Ref. [43] lacks its DOI 10.1007/BF01218021.
19- Ref. [77] seems to be published, see https://doi.org/10.1142/9789814618144_0010.
Recommendation
Ask for minor revision
Report
The authors have considered the suggestions of the referees on a very serious level and improved the presentation and results of their manuscript. Notably, the comparison to other methods is now very useful for the general reader. Publication as a SciPost article is now well supported.
Recommendation
Publish (surpasses expectations and criteria for this Journal; among top 10%)
Strengths
The authors have significantly updated their manuscript with detailed DMRG simulations of the magnetization curve of the ATLAFM and added detailed comparisons with several existing numerical cluster-based methods. The good agreement between DMMFT and their DMRG simulations strongly supports the central claim of the manuscript.
Report
The resubmitted manuscript meets SciPost criteria. I recommend its publication.
Recommendation
Publish (surpasses expectations and criteria for this Journal; among top 10%)

---

## Round 2 · Author Response

We write to resubmit our revised manuscript titled “Density-Matrix Mean-Field Theory” for your consideration in SciPost Physics.
In response to the concerns raised in the previous round of review, a primary focus was placed on evaluating the accuracy of our method. Taking the advice from the referees, we have incorporated a benchmark calculation using the state-of the-art technique density-matrix renormalization group (DMRG). Additionally, we have expanded our manuscript to include more comprehensive and quantitative comparisons with both DMRG results and results reported in previous literatures, as suggested by one of the referees. We find a good alignment between our results and those obtained through DMRG, as well as with previously reported results, which provides more robust support for the accuracy and reliability of our proposed method.
Moreover, we have expanded our discussions by comparing our method not only to dynamical mean-field theory (DMFT) and DMRG, but also to linked-cluster expansion (LCE) and cluster variation method (CVM). These addresses suggestions made by referee 3 to connect our method to alternative cluster-based approaches. Furthermore, we have also included an Appendix that provides detailed implementation guidelines for our method applied to the antiferromagnetic Heisenberg model on triangular lattices (AFHTL). This addition directly addresses the concerns raised by referee 1.
In addition, we have expanded our discussions about the Affleck-Kennedy-Lieb-Tasaki (AKLT) model and the AFHTL. Regarding the AKLT model, while we acknowledge its simplicity, the capability of our method to detect the Haldane phase with topological order while the conventional Néel order vanishes provides valuable insights beyond known mean-field methods. Furthermore, while we have presented our results for the special case of β=1/3, our conclusions are still valid for the generic range of 0<β<1/3, where the topological phase persists, since our calculations do not rely on special properties at β=1/3. Additionally, our work is not restricted to the demonstrative examples discussed in the manuscript. It can serve as an efficient tool for studying various quantum orders, including, for example, stripy phases in J1-J2 Heisenberg model and dimerized phases. Providing references to relevant works will enhance the understanding of readers with broader interests.
In conclusion, this revised manuscript comprehensively addresses all essential points raised by the referees, and provides substantial evidence supporting the accuracy and efficacy of our method. We are confident that these enhancements align with the high standards of your flagship journal, SciPost Physics.
We appreciate your editorial efforts and thank you for considering our manuscript for publication in your esteemed journal.
Sincerely,
Junyi Zhang,
on behalf of the authors.

---

## Round 2 · List of Changes

1. To address the referees’ major concern about the accuracy of our method, we have conducted additional benchmark calculations for the antiferromagnetic Heisenberg model on triangular lattices (AFHTL) using the state-of-the-art density-matrix renormalization group (DMRG) technique. The comparison between our results and those obtained through DMRG, as well as with previously reported results, shows good alignment, which provides robust support for the accuracy and reliability of our proposed method.
a) Page 12-Page 14, since the last paragraph on Page 12 through the end of Sec. 3.
We have expanded Sec. 3.2 to include detailed DMRG calculations for the AFHTL and relevant discussions. This includes descriptions of the system geometry used in DMRG calculations and the magnetization curve obtained. Additionally, we have provided quantitative comparisons between the results of DMRG and our method, density-matrix mean-field theory (DMMFT). We have also compared our results to those from previous reports suggested by Referee 2.
b) Page 13, Fig. 3.
we have included a new figure (Fig. 3) illustrating the system geometry used in the DMRG calculations, along with overlaid magnetization curves calculated using DMMFT, conventional mean-field theory, and DMRG.
2. As suggested by Referee 3, we have discussed the connection and difference of our method to other cluster-based methods, the cluster variation method (CVM) and the linked-cluster expansion (LCE)
a) Page 14 – Page 17, Sec. 4.1.
We have reorganized Section 4.1 to provide a more structured comparison between DMMFT and other methods. The original Section 4.1 has been split into two subsections: 4.1.1 and 4.1.2, dedicated to comparisons with DMRG and DMFT, respectively. Additionally, two new subsections, 4.1.3 and 4.1.4, have been added to compare DMMFT with CVM and LCE, respectively. These additions provide a more comprehensive analysis of the similarities and differences between our method and alternative cluster-based approaches.
b) Page 3, the third paragraph.
We have also updated the third paragraph on Page 3 to introduce the new discussions related to CVM and LCE in Section 4.1. This introductory paragraph motivates the connection of our method to alternative cluster-based methods, providing readers with a clear understanding of the context for the subsequent discussions.
3. In response to Referee 3’s inquiry regarding the translational symmetry, we have added a new paragraph to discuss this issue explicitly. Particularly, we have provided a detailed protocol for imposing lattice symmetries, including translational symmetries, with Eq. 25 in the revised manuscript. This addition ensures a consistent treatment of symmetries throughout our method.
a) Page 16, last paragraph of 4.1.2.
4To address Referee 1’s request “a simple illustration of it on some simple toy model would help to understand it better”, we have included a new appendix section, Append. B. This appendix provides a detailed description of the implementation of DMMFT for the AFHTL, along with a step-by-step pseudocode. This addition aims to enhance the understanding of our method through a more accessible example.
a) Page 20-23, Apped. B.
5. We appreciate Referee 2 for bringing to our attention some relevant previous studies that were previously overlooked. These studies have been properly cited in our revised manuscript (Ref. 47-50). Additionally, we have provided precise citations to describe the ground states of the AKLT model, as requested by Referee 2. We have expanded our discussion to justify the non-trivial insights provided by the AKLT model and to explain the relevance of the J1-J2 Heisenberg model to our method and to readers with broader interests.
a) Page 3, last paragraph, Line 7.
We have updated the references regarding the AFHTL according to Referee 2’s suggestion, now citing Ref. 47-50 in the revised manuscript.
b) Page 8, the introductory paragraphs of Sec. 3.
We have expanded the discussion to justify the significance of the AKLT model and to explain the relevance of the J1-J2 Heisenberg model. This ensures that our readers understand the broader context and importance of these models in relation to our method.
c) Page 9, third to the last paragraph.
We have revised statement concerning the ground state properties of the AKLT model as “The AKLT model is an exactly solvable system, showcasing non-trivial topological order. Notably, it features spin-1/2 (fractional to spin-1) edge states and exhibits a 4-fold degeneracy of the ground states for an open chain [43], which persists even for β < 1/3 deviating from its integrable point, provided the gap does not close [60, 68].”, where we have provided citations (Ref. 43, 60, 68) to support the statement. The statement about the nature of the ground states of the AKLT model has been made by AKLT in their original paper, here cited as Ref. 43. Moreover, it is important to recognize that the properties of these topological states do not depend on the special point of β = 1/3; instead, they persist within the range of 0 < β < 1/3, supporting the existence of topological ground states. This aspect justifies our choice of the AKLT model and ensures the generality of our discussions. The statement regarding the persistence of topological states can be found and is cited as Ref. 60 and 68.
6. We conducted a thorough review of the references and made necessary corrections to ensure accurate formatting of all chemical compounds, names of individuals, and model names. Specifically, we have overridden the default compilation bib style to rectify misprinted titles, such as "XXZ," and names like "Néel" in the titles, ensuring their proper formatting.

---

## Editorial Decision

unknown